# Towards the Effect of Large Language Models on Out-Of-Distribution Challenge in Text-Attributed Graphs

## Abstract

Text-Attributed Graphs (TAGs), where each node is associated with text attributes, are ubiquitous and have been widely applied in the real world. The Out-Of-Distribution (OOD) issue, i.e., the training data and the test data not from the same distribution, is quite common in learning on real-world TAGs, posing significant challenges to the effectiveness of graph learning models. Recently, Large Language Models (LLMs) have shown extraordinary capability in processing text data, and have demonstrated tremendous potential in handling TAGs. However, there is no benchmark work that systematically and comprehensively investigates the effect of these LLM-based methods on alleviating the OOD issue on TAGs. To bridge this gap, we first develop OOD-TAG, a comprehensive OOD benchmark dataset in TAGs which consists of diverse distributions. Meanwhile, we conduct a systematic and comprehensive investigation on OOD-TAG with different LLM pipelines for graphs. In addition, we provide original observations and novel insights based on the empirical study, which can suggest promising directions for the research of LLMs in addressing the OOD challenges on TAGs. Our code and dataset are available in https://anonymous.4open.science/r/GraphOOD-benchmark-5FCF/.

## 1 Introduction

Graph typically consists of a bunch of nodes denoting different entities and numerous edges describing the connections among these nodes. In many real-world graphs, nodes are often associated with textural attributes, such as paper citation networks, where each paper node is described with its title and abstract (McCallum et al., 2000; Sen et al., 2008), and recommendation graphs, where both the user and item nodes are associated with characteristic descriptions in text (Zhu et al., 2021). These graphs along with textual node attributes are usually denoted as text-attributed graphs (TAGs), and they are ubiquitous in various domains. Graph neural networks (GNNs) are effective extensions of deep neural networks on graphs, and have shown impressive ability in graph representation learning. Despite great success achieved in diverse graph-related tasks, most GNNs assume that the training and test data are from the same distribution. However, such assumption is often violated in real-world graph applications. For examples, the paper citation connections can shift with the change of academic hot spots, and items in recommendation graphs can update with the process of science and technology. These shifts in graphs have been empirically and theoretically proven to be harmful for the performance of GNNs in graph-related tasks (Zhu et al., 2021) and many research efforts are made to address this so-called OOD (Out of Distribution) challenge (Gui et al., 2022; Yang et al., 2021) in graphs.

Recently, large language models(LLMs) have shown extraordinary context-aware knowledge and semantic understanding ability (Brown, 2020). Pioneers such as GPT-4 (Achiam et al., 2023) and Claude-3 (Anthropic, 2024) have achieved revolutionary success in various applications from different domains. Given the huge progress LLMs have made in handling the texture-related tasks and great potential of LLMs in dealing with TAGs (Zhu et al., 2021; Zhao et al., 2022; He et al., 2023a; Chien et al., 2021), a natural question arises: *Can LLMs bring a new dawn for the OOD challenge in the graph domain?*

To answer this question, we embark upon a preliminary study by systematically conducting a series of empirical investigations. Specifically, we benchmark three widely-received LLM-compatible

pipelines (Chen et al., 2024; 2023; Zhang et al., 2024; He et al., 2023b;a). in graph OOD challenge, including (1) *LLMs-as-Enhancers*: LLMs are utilized to enhance the input node features, either at the text level or the embedding level; (2) *LLMs-as-Annotators*: LLMs are adopted to make predictions for some nodes; (3) *LLMs-as-Predictors*: LLMs are used to make direct predictions for all the nodes. In the above pipelines, the first two are GNN-integrated pipelines where GNNs are taken to make the final predictions with assistance of LLMs, while the last one is GNN-excluded, where we directly leverage LLMs to make the final predictions. In addition, we develop an OOD dataset, OOD-TAG, including a series of widely-received protocols for distribution shift, which includes four OOD types on five TAG datasets. 16 methods from three pipelines(9 from *LLMs-as-Enhancers*, 2 from *LLMs-as-Annotators* and 5 from *LLMs-as-predictors*) are examined on OOD-TAG based on several aspects, including performance on both the in-distribution (ID) test set and the OOD test set; (2) Performance under various OOD types; (3) Expenditure for LLM usage in different pipeline implementations. We have conducted comparisons among the empirical results and these methods with in-depth analysis and discussion. Our contributions of this work are summarized as follows:

**An OOD dataset on TAGs.** We develop OOD-TAG, which consists of four OOD types among five TAG datasets, and is flexible to include more OOD types on additional TAGs.

**Comprehensive and reproducible study.** We conduct a systematical study on 16 methods with three LLM pipelines on OOD-TAG, and compare their performance from different perspectives. Both the baseline codes and OOD-TAG are available at https://anonymous.4open.science/r/GraphOOD-benchmark-5FCF/.

**Insights.** Through a comprehensive study, we have concluded several findings: (1) Using LLMs as feature-level enhancer or annotators can help significantly alleviate the OOD challenge in TAGs with reasonable expenses; (2) When serving as a predictor, LLMs show more promising generalization performance after fine-tuning; (3) Well-designed prompts that can efficiently capture the graph information are essential for the LLMs' performance.

## 2 BACKGROUND

In this section, some preliminaries of our work will be provided. First, we will give a brief introduction about Text-Attributed-Graphs (TAGs) and the node classification problem on TAGs. Next, the Out-Of-Distribution (OOD) challenge on TAGs will be illustrated. In addition, we will discuss some basic concepts and recent advances on Large Language Models(LLMs) on graphs, separately.

**Text-Attributed-Graphs(TAGs).** A text-attributed graph can be formulated as $G = (V, E)$, where $V = \{v_1, ..., v_N\}$ constitutes the set of $N$ nodes and $E = \{e_1, ...e_k\}$ is the edge set describing the connection among these nodes. Specifically, for each node, there is an associated text description $s_i$. TAGs are prevalent in real-world applications. To take OGBN-ARXIV (Hu et al., 2021b) as an illustrative example, each node in this citation graph represents an academic paper with a text description consisting of its title and abstract, and each edge denotes a citation relationship between two papers.

**Node Classification on TAGs.** In this study, we mainly focus on the node classification problem on TAGs. Specifically, given a TAG $G = (V, E)$ with a set of note attributes $S = \{s_1, ..., s_N\}$, where the node set $V$ can be divided into a labeled node set $V_L$ and an unlabeled node set $V_U$, the aim of node classification is to predict the labels of nodes from $V_U$. Let's continue with the example of OGBN-ARXIV. The label of each paper node is the domain category it belongs to, such as computer vision and natural language processing.

**Out-Of-Distribution(OOD) challenges on TAGs.** Out-of-distribution challenges typically refer to scenarios where the distributions of training set and test set are different. It is very common in a variety of applications and has been widely recognized as a significant reason for the degradation of model performance (Hand, 2006; Tzeng et al., 2017; LEARNING). The OOD issue is prevalent and intricate in TAGs, which can manifest in diverse perspectives. For example, in OGBN-ARXIV, the distribution shift between the training set and the test set can happen in the aspect of the publication periods or the research directions of these papers.

**Large Language Models (LLMs) on Graphs.** Large Language Models (LLMs) have exhibited superior performance in various applications (Zhu et al., 2021; Zhao et al., 2022; Bubeck et al., 2023; Chien et al., 2021) and have demonstrated extraordinary potential in Text-Attributed Graphs (Liu et al., 2023; Gao et al., 2023; Hu et al., 2020b; Yasunaga et al., 2022). There are three popular

pipelines for applying LLMs to TAGs, including *LLMs as enhancers*, *LLMs as predictors* and *LLM as annotators*. For LLMs as enhancers, LLMs are leveraged as enhancer for node representations. Specifically, we can either take advantages of LLMs to enrich the text descriptions of nodes, or generate LLM-based embeddings given the original text attributes. In addition, it is also common to directly leverage LLMs as a predictor via prompting. Although being effective and straightforward, this method comes at a huge cost. To ameliorate the issue, a compromise solution has been proposed to use LLMs as annotators. In other words, LLMs are used to make predictions for a specific bunch of samples, and then such information is leveraged to enhance some small-scale models, such as GNNs, which are used for the final prediction. For the clear description of LLMs in different pipelines, we also roughly divide LLMs into two categories based on whether users have access to their embeddings following Chen et al. (2024). The embedding-visible LLMs, denoted as $LLM^*$ in this paper, ref to LLMs that enable users to get embeddings for specific text inputs, such as *BERT* (Devlin et al., 2019) and *Deberta* (He et al., 2021b). The embedding-invisible LLMs only offer text-based user interface, and make both embeddings and parameters invisible to users, such as ChatGPT , which is deployed as a web service.

## 3 BENCHMARK DESIGN

To conduct a comprehensive study on the effect of LLMs on the graph OOD challenge, we build up the OOD-TAG dataset and systematically explore the three LLM pipelines mentioned above, also to bridge the gap that there is currently no such systematic OOD dataset in TAGs. In this section, a detailed description of the OOD-TAG dataset built by us will first be provided. Then, we will further elaborate the pipelines for LLMs in graphs studied in this benchmark.

### 3.1 DATASETS

To build up the OOD-TAG dataset, we consider five widely-used datasets in graph-related research: CORA (McCallum et al., 2000), PUBMED (Sen et al., 2008), CITESEER (Giles et al., 1998), WIKICS (Mernyei & Cangea, 2022) and ARXIV (Hu et al., 2021b). All of the five datasets are citation datasets. These text-attributed graphs provide original textual sentences and extract node attributes from textual sources such as paper titles and abstracts. More details of datasets are include in Appendix A.1.

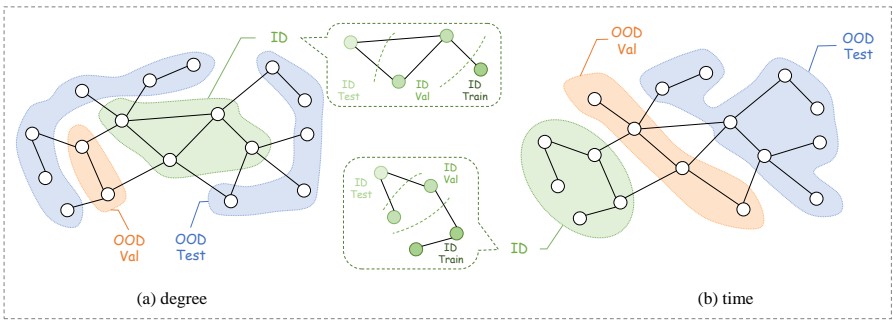

Figure 1: Two illustrative examples of covariate shift. (a) demonstrates the dataset split based on degree as the criterion for domain split, with different colors indicating the training, test, and validation sets. Nodes with high degrees are designated as the training set, while those with the relatively low degrees are assigned to the test set. (b) presents a scenario in which time is used as the split criterion. As the citation network evolves over time, nodes in the graph are categorized accordingly. Specifically, the ID part comprises data from 2005 to 2007, the validation part includes papers from 2011 to 2014, and the test set consists of papers published between 2018 and 2020.

Following GOOD (Gui et al., 2022), we adopt the principles of covariate shift and concept shift to develop the OOD data splits. A description of these two shifts is given in the Appendix A.2. Specifically, for CORA, PUBMED, CITESEER and WIKICS, two domain criterion are considered, named word and degree, which reflect node features and graph structure, respectively. The word reflects diversity of words in a paper, and the degree demonstrates the connection of node with its neighbours. For ARXIV. we employed degree and time as criterion for domain split. Specifically, for

the time shift, we take papers published between 2018 to 2020 for test, while those published in other time periods for training. Based on these diverse split principles, we develop four Out-of-Distribution (OOD) datasets for each original dataset. More specific details in Table 8 in Appendix A.1. Two illustrative examples of covariate shift are shown in Figure 1.

## 3.2 PIPELINES FOR LLMs IN GRAPHS

Three popular pipelines for applying LLMs to TAGs are considered in this benchmark, including *LLMs as enhancers*, *LLMs as predictors* and *LLMs as annotators*. Figure 2 provides the overall framework of these three pipelines.

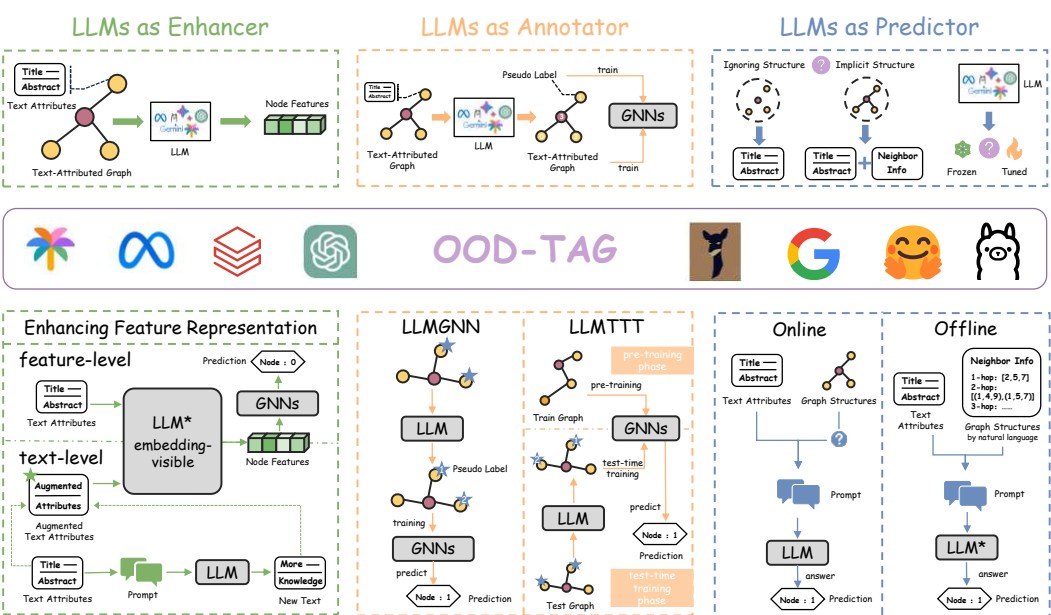

Figure 2: The pipelines for applying LLMs to TAGs. It is designed with three parts: (1) *LLMs as Enhancer:* LLMs are used to enhance feature representations in feature and text level. (2) *LLMs as Annotator:* Two different methods are used LLMs as annotators to assist in training GNN models. (3) *LLMs as Predictor:* Online and Offline LLMs are directly independently perform predictive tasks on graph structures. In all figures, we use "LLM*" to denote embedding-visible LLMs, and "LLM" to denote embedding-invisible LLMs.

In the pipeline of *LLMs as enhancers*, LLMs are used to enhance the node features. There are two mainstream ways to leverage LLMs to enrich the text attributes, *i.e.*, feature-level enhancement and text-level enhancement (Chen et al., 2023). The feature-level enhancement utilizes LLMs' knowledge by encoding text attributes in to features. The text-level enhancement injects the LLMs' knowledge by augmenting the text attributes via extended text information. Specifically, for the feature-level enhancement, the LLMs work as feature embedding models whose outputs are then fed into small-scale backbone models, such as GNNs. As illustrated in Figure 3 in Appendix B, only embedding-visible LLMs are taken into account to encode the text information. For the text-level enhancement, embedding-invisible LLMs are used to enrich the text attributes via various prompting strategies. Recently, several papers (He et al., 2023b; Chen et al., 2024) have explored the text-level enhancement in different manners. Here, we take TAPE (He et al., 2023b) and KEA (Chen et al., 2024) as baseline methods and the illustrative examples of these two augmentations are shown in Figure 4 in Appendix B.2. More details about KEA and TAPE are included in Appendix B.2.

While LLMs are effective predictors, their usage usually comes with high costs and slow rate. To address these challenges , some research (Chen et al., 2024; 2023) propose to leverage the zero-shot inference capacity of LLMs for training smaller models, such as GNNs. To provide a more comprehensive survey of various LLM pipelines, we examine two approaches that utilise *LLMs as annotators* and evaluate their performance on the OOD-TAG dataset. These two approaches are further illustrated in Figure 5 in Appendix C. First, we conduct an evaluation of label-free node

classification on graphs using LLMs (LLMGNN (Chen et al., 2024)). Specifically, LLMs are used to annotate a selected set of nodes, following which GNNs are trained using the annotations provided by LLMs to predict for the remaining numerous nodes. We have also conducted an evaluation of the test-time training pipeline, LLMTTT (Zhang et al., 2024), which performs test-time adaptation based on annotations provided by LLMs for a carefully-selected node set. In particular, LLMTTT incorporates a hybrid active node selection strategy that takes into account not only node diversity but also prediction signals from the pre-trained model. With annotations from LLMs, a two-stage training strategy is devised to fine-tune the test-time model using the limited and noisy labels.

In the *LLMs-as-Predictors* pipeline, LLMs are directly leveraged to solve the graph-related tasks. There are two ways to utilize LLMs on graph tasks. One is to directly use the pre-trained LLMs, while the other one is to first fine-tune the pre-trained LLMs on the specified graph datasets, and then to use the fine-tuned LLMs to make predictions. In order to comprehensively assess the impact of using LLMs as predictors on the OOD challenge in TAGs, both pre-trained and fine-tuned LLMs are evaluated. These two approaches are further illustrated in Figure 6 in Appendix D. For the online LLMs, we employ ChatGPT (Bahrini et al., 2023) for our experiments. Specifically, we leverage different prompts to interact with LLMs, including the prompts only consisting of node attributes, and the prompts containing both the node attributes and the structure information. For the offline LLMs, we choose InstructGLM (Ye et al., 2024), which employs LLama (Radford et al., 2018; Hu et al., 2021a) as its backbone and has undergone fine-tuning on a number of graph benchmark datasets. InstructGLM is designed to generate highly scalable prompts based on natural language instructions. It utilises natural language to describe the geometric structure and node features of graphs, thereby enabling the tuning of an LLM for learning and inference tasks on graphs in a generative manner.

## 4 EXPERIMENTS

In this section, we systematically investigate the potential of LLMs in alleviating the Out-Of-Distribution challenge on TAGs. Specifically, we aim at answering the following questions, where the first three are to use LLMs as assistance of GNNs, while the last one is to use them as final predictors:

**Q1:** Can LLMs help alleviate the OOD challenge on TAGs via enhancing the node embeddings?
**Q2:** Can LLMs help alleviate the OOD challenge on TAGs via enhancing the node text attributes?
**Q3:** Can LLMs help alleviate the OOD challenge on TAGs via annotating some test nodes?
**Q4:** Can LLMs help alleviate the OOD challenge on TAGs via directly working as predictors?

### 4.1 LLMS-AS-ENHANCERS

#### 4.1.1 FEATURE-LEVEL ENHANCEMENT

**Experiment Implementation.** To answer **Q1**, we implement four typies of LLMs as feature enhancers, including **(1) Fixed PLMs and LLMs**: DeBERTa (He et al., 2021a) and LLaMA (Touvron et al., 2023), both of which are embedding models that encode each word as a fixed-length vector. **(2) Local Sentence Embedding Models:** Sentence-BERT (Reimers & Gurevych, 2019) and e5-large (Wang et al., 2022), which are designed to encode sentences into fixed-length vectors, with the overall aim of making the embeddings of similar sentences close to each other in the latent space. **(3) Online Sentence Embedding Models:** The text-ada-embedding-002 model, which operates as an online sentence embedding model. Meanwhile, **(4) the non-contextualized shallow embeddings models**: TF-IDF and Word2Vec (Mikolov et al., 2013). TF-IDF is a statistical language model that trains the encoder on the entire dataset. Word2Vec generates shallow embeddings using a relatively shallow neural network that learns from contextual information. Regarding the small-scale backbone models, graph convolutional networks (GCN) (Kipf & Welling, 2017), and multi-layer perceptrons (MLP) (Tolstikhin et al., 2021) are employed to deal with the TAGs enhanced by the LLMs.

**Results.** As shown in Table 1 and 2, the gap between in-distribution (ID) and out-of-distribution (OOD) performance is notably larger for concept degree, ranging from 4% to 12%, while for concept word, the difference remains below 3%. This demonstrates that the distribution shift in terms of graph structure can cause a severer impact on the performance of GCN than MLP, while the performance gap of MLP is more significant for the distribution shift concerning word than that concerning degree, as illustrated in Table 11, 12 and Figure 7 in Appendix E, which reveals the message passing mechanism in GCN can help mitigate the influence caused by feature distribution.

Table 1: The results of LLMs as feature-level enhancers for concept degree shift. Different GCN models are trained with different embeddings on the ID training set. "ID" denotes the GCN performance (%) on the ID test set, and "OOD" denotes the GCN performance (%) on the OOD test set. "Avg Rank" is the average ranking of this method among all the methods across various datasets. For each kind of features, we use yellow to denote the best performance, green the second best one, and blue the third one.

| | Cora | | Pubmed | | CiteSeer | | Wikics | | ArXiv | | Avg Rank | |
|---|---|---|---|---|---|---|---|---|---|---|---|---|
| | ID | OOD | ID | OOD | ID | OOD | ID | OOD | ID | OOD | ID | OOD |
| *Non-contextualized Shallow Embeddings* | | | | | | | | | | | | |
| TF-IDF | 88.36 | 84.76 | 90.20 | 82.15 | 74.15 | 63.12 | 84.72 | 75.96 | 76.06 | 64.76 | 3.8 | 3 |
| Word2Vec | 88.56 | 83.06 | 89.54 | 73.39 | 64.77 | 61.89 | 83.29 | 75.36 | 75.72 | 62.00 | 5.2 | 5.2 |
| *Local Word Embedding models* | | | | | | | | | | | | |
| DeBERTa | 82.69 | 79.19 | 86.58 | 66.59 | 52.85 | 47.18 | 71.49 | 60.04 | 60.69 | 43.69 | 7 | 7 |
| LLaMa | 87.06 | 80.48 | 88.73 | 75.23 | 79.00 | 69.52 | 83.20 | 72.35 | 76.78 | 65.68 | 4.2 | 4.6 |
| *Local Sentence Embedding Models* | | | | | | | | | | | | |
| SBERT | 91.54 | 83.73 | 90.34 | 79.13 | 77.31 | 68.17 | 84.00 | 74.69 | 76.37 | 66.41 | 2.4 | 3.2 |
| E5 | 90.05 | 83.08 | 91.17 | 81.10 | 76.77 | 71.78 | 83.58 | 74.35 | 75.97 | 65.11 | 3.2 | 3.4 |
| *Online Sentence Embedding Model* | | | | | | | | | | | | |
| Ada | 89.05 | 86.98 | 90.72 | 80.98 | 78.54 | 74.43 | 83.79 | 77.19 | 77.46 | 66.25 | 3.2 | 1.6 |

Table 2: The results of LLMs as feature-level enhancers for concept word shift. Different GCN models are trained with different embeddings on the ID training set. "ID" denotes the GCN performance (%) on the ID test set, and "OOD" denotes the GCN performance (%) on the OOD test set. "Avg Rank" is the average ranking of this method among all the methods across various datasets. For each kind of features, we use yellow to denote the best performance under a specific GCN model, green the second best one, and blue the third one.

| | Cora | | Pubmed | | CiteSeer | | Wikics | | ArXiv | | Avg Rank | |
|---|---|---|---|---|---|---|---|---|---|---|---|---|
| | ID | OOD | ID | OOD | ID | OOD | ID | OOD | ID | OOD | ID | OOD |
| *Non-contextualized Shallow Embeddings* | | | | | | | | | | | | |
| TF-IDF | 89.35 | 87.11 | 87.86 | 87.85 | 70.41 | 71.44 | 84.17 | 81.97 | 75.11 | 71.50 | 3 | 3.2 |
| Word2Vec | 85.97 | 85.02 | 84.55 | 85.32 | 63.54 | 68.45 | 81.78 | 80.80 | 73.58 | 69.75 | 6 | 6 |
| *Local Word Embedding Models* | | | | | | | | | | | | |
| DeBERTa | 83.29 | 81.36 | 82.69 | 83.27 | 55.13 | 45.39 | 72.47 | 69.48 | 55.28 | 50.18 | 7 | 7 |
| LLaMa | 87.45 | 85.82 | 86.42 | 85.73 | 72.77 | 75.56 | 83.4 | 81.30 | 75.53 | 71.53 | 4 | 3.6 |
| *Local Sentence Embedding Models* | | | | | | | | | | | | |
| SBERT | 87.97 | 85.78 | 87.14 | 86.78 | 73.36 | 75.56 | 83.46 | 81.68 | 75.94 | 71.02 | 3 | 3.8 |
| E5 | 88.83 | 86.51 | 88.30 | 88.39 | 73.36 | 74.87 | 83.28 | 81.00 | 76.68 | 71.58 | 2.6 | 3 |
| *Online Sentence Embedding Model* | | | | | | | | | | | | |
| Ada | 87.36 | 87.25 | 87.74 | 87.93 | 77.93 | 78.99 | 84.51 | 82.89 | 75.71 | 71.75 | 2.2 | 1.2 |

TF-IDF demonstrates superior performance in two *Non-contextualized Shallow Embeddings*. TF-IDF outperforms Word2Vec across different datasets in concept shift of degree, with performance on Cora and Wikics being roughly equivalent, while TF-IDF exhibited a slight advantage of approximately 2% on other datasets. Across all datasets in the concept degree category, the ID and OOD performance indicated that TF-IDF consistently exceeded Word2Vec by 2% to 7%. This performance discrepancy may be attributed to the nature of TF-IDF as a statistical language model, which trains the encoder specifically on each dataset, while the utilized Word2Vec model is pre-trained on a fixed dataset (GoogleNews-vectors-negative300).

Among the LLM models evaluated, DeBERTa exhibited the poorest performance across all datasets in various data shift scenarios. As for LLaMa, its performance is not consistent in different cases. It performs worse than Word2Vec on Cora and Wikics in the case of concept degree shift, both in terms of accuracy and generalization, while exhibiting slightly better generalization performance than Word2Vec on the PubMed and ArXiv datasets, and significant improvement over TF-IDF and Word2Vec on the Citeseer dataset .

The local sentence embedding E5 and SBERT methods exhibit comparable performance on ID and OOD datasets and gain accuracy improvements when compared to Word2Vec. While they show a slight improvement over TF-IDF on the ID dataset, the performance on the OOD dataset is generally inferior to TF-IDF, with the exception of a notable improvement on the Citeseer dataset. When

compared to Word2Vec, although the impact on ID accuracy in the PubMed and ArXiv datasets is minimal, there exists a significant enhancement in generalization performance, with both ID and OOD accuracy showing considerable increases on the Citeseer dataset. The online sentence embedding Ada's performance on ID dataset is comparable to that of TF-IDF. However, on OOD datasets, Ada consistently shows improvements compared to TF-IDF, with a particularly noticeable enhancement on the Citeseer dataset. Comprehensive results concerning LLMs as feature-level enhancer are in Appendix E.

**Observation 1:** *Online sentence embedding model, represented by Ada, outperforms other embedding methods consistently in both the ID test and OOD test. Local sentence embedding models also show relatively impressive performance, while the local word embedding models demonstrate unstable performance. In addition, TF-IDF achieves surprisingly satisfying performance in the OOD scenarios.*

### 4.1.2 TEXT-LEVEL ENHANCEMENT

**Experiment Implementation.** To answer **Q2**, we use the embedding-invisible LLM gpt-3.5-turbo-0613 to provide the text-level knowledge and the embedding-visible LLM e5-v2-large to encode the texts into embeddings. MLP (Tolstikhin et al., 2021) are employed as the backbone. The results of original texts, predictions, explanation, KEA-I texts and KEA-S texts are denoted as "TA", "P", "E", "KEA-I" and "KEA-S", respectively. The results of ensemble methods are represented by their components. For example, the "TA+P" result denotes the ensemble prediction combined from the prediction matrices from the GNNs trained on the original texts and prediction texts of TAPE. Considering the huge cost to use LLMs, we only conduct experiments on the Cora and Pubmed datasets.

**Results.** As shown in Table 3, "E","TA+E" and "TA+P+E" from TAPE demonstrate consistent improvements over the "TA" baseline. "P" shows significantly inferior performance compared to other methods, while "E" brings performance gain compared to the baseline "TA", which is especially obvious in the Pubmed dataset, suggesting that the effectiveness of TAPE mainly come from the explanations (E). Furthermore, the ensemble methods, inluding "TA+E" and "TA+P+E" indicate consistent performance improvements across different methods.

KEA exhibits better generalization performance than the baseline method TA, while showing overall inferior performance compared to TAPE, according to Table 3. However, KEA achieves better performance than TAPE in Cora than Pubmed, especially in covariant shift shown in Table 17, 18and 19 in Appendix F. We analyse the performance discrepancy can be partially attributed to the accuracy difference of pseudo label between different datasets. When the pseudo label accuracy of LLMs is high, such as in Pubmed, TAPE can get significant benefits since its "P" and "E" are based on the pseudo label. However, when the prediction of LLM is not accurate, as in Cora, KEA can help mitigate the impact of erroneous predictions, leading to better generalization. This highlights the complementary strengths of both methods in different scenarios, with TAPE excelling in accuracy and KEA providing robustness against prediction errors.

The accuracy of the pseudo labels (i.e. similar as the zero-shot predictor in the following section) is parsed directly from TAPE's LLM responses. As shown in Table 3, the accuracy of the pseudo labels reaches 93–94% in the Pubmed dataset, surpassing the TA (E5) method by approximately 5%. In contrast, in the Cora dataset, the accuracy is about 64–67%, which is lower than the TA (E5) method by approximately 15–25%. In other words, when TAPE converts the pseudo labels into a ranked prediction matix embedding and then trains the GCN with the embeddings, both the accuracy and generalization performance do not match those achieved by directly utilizing the pseudo labels or the backbone models directly trained on the baseline embedding. In the easy task, such as the three-class classification task in Pubmed, it is much better and simpler to directly use the pseudo label prediction from LLMs, than to train a model with the embeddings encoded from the predictions.

**Observation 2:** *Using LLMs as text enhancer can improve the backbone models' performance on the OOD challenge. TAPE shows relatively consistent improvements across various scenarios, with main performance gain coming from the explanation part. KEA demonstrates promising result in relatively complex task (such as Cora, where there are seven classes that exist overlaps), where the direct prediction from LLMs is far from satisfactory. Meanwhile, the ensemble method to combine different texts exhibits consistent superiority.*

Table 3: The results of LLMs as text-level enhancers. Different GCN models are trained with different texts on the ID training set. "ID" denotes the GCN performance (%) on the ID test set, and "OOD" denotes the GCN performance (%) on the OOD test set. "Avg Rank" is the average ranking of this method among all the methods across various datasets. "Pseudo Label" is the prediction accuracy coming from the direct prediction from LLMs during the TAPE process. For each kind of methods, we use yellow to denote the best performance, green the second best one, and blue the third one.

| Concept Degree | Cora | | Pubmed | | Avg Rank | | Concept Word | Cora | | Pubmed | | Avg Rank | |
|---|---|---|---|---|---|---|---|---|---|---|---|---|---|
| | ID | OOD | ID | OOD | ID | OOD | | ID | OOD | ID | OOD | ID | OOD |
| TA(E5) | 90.85 | 83.44 | 89.88 | 80.99 | 5 | 6.5 | TA(E5) | 83.08 | 83.33 | 87.36 | 87.51 | 6.5 | 6 |
| Pseudo Label | 66.17 | 66.55 | 94.66 | 93.52 | 5.5 | 5 | Pseudo Label | 67.85 | 64.13 | 93.58 | 93.58 | 5 | 5 |
| P | 68.56 | 62.70 | 78.56 | 69.52 | 9.5 | 10 | P | 64.86 | 58.37 | 84.12 | 83.90 | 10 | 10 |
| E | 90.35 | 83.68 | 91.62 | 85.84 | 3 | 4 | E | 84.49 | 82.30 | 92.03 | 91.70 | 2 | 4.5 |
| TA+E | 91.04 | 84.23 | 91.32 | 84.24 | 2.5 | 3 | TA+E | 84.49 | 84.90 | 90.78 | 90.61 | 3 | 3.5 |
| TA+P+E | 90.55 | 84.18 | 91.33 | 84.12 | 3 | 4 | TA+P+E | 85.05 | 84.52 | 90.88 | 90.98 | 2 | 2.5 |
| KEA-I | 84.98 | 82.27 | 90.29 | 80.00 | 7.5 | 8 | KEA-I | 81.68 | 82.76 | 86.78 | 87.36 | 7.5 | 7 |
| KEA-S | 87.96 | 84.40 | 88.93 | 77.36 | 8 | 5.5 | KEA-S | 81.59 | 81.47 | 85.67 | 87.22 | 8.5 | 8.5 |
| KEA-I+TA | 88.06 | 84.13 | 90.46 | 81.36 | 5.5 | 5 | KEA-I+TA | 84.30 | 84.62 | 87.48 | 87.97 | 5 | 3.5 |
| KEA-S+TA | 88.86 | 85.35 | 90.32 | 80.86 | 5.5 | 4 | KEA-S+TA | 83.55 | 84.18 | 87.56 | 88.29 | 5 | 4.5 |

## 4.2 LLMs-as-Annotators

**Experiment Implementation.** To answer **Q3**, we implement two methods leveraging LLM knowledge to train GNNs, essentially using LLMs as annotators. The first one is to train a GNN model from scratch with the annotations from LLMs on partial test nodes, termed LLMGNN. The other one is denoted as LLMTTT, which finetunes a pre-trained GNN model with the annotations from LLMs. In both methods, the backbone GNN model is the graph convolutional network (GCN) model, and the GPT-3.5-turbo-0613 model is used to generate pseudo labels. The number of annotated test nodes is set as the value of 20 times the number of node categories in LLMGNN, and that in LLMTTT is 10% of the number of test samples.

Table 4: The results of LLMs as annotator. "GCN" is the performance (%) of a well-trained GCN on OOD test dataset. The embeddings used by GCN are produced by sbert. "LLMTTT" and "LLMGNN" are performance (%) of two different pipelines in which LLMs are used as annotators.

| | concept_degree | | | | | concept_word | | | | |
|---|---|---|---|---|---|---|---|---|---|---|
| | Cora | Pubmed | CiteSeer | Wikics | ArXiv | Cora | Pubmed | CiteSeer | Wikics | ArXiv |
| GCN (sbert) | 86.50 | 80.32 | 68.60 | 72.67 | 64.69 | 87.48 | 87.34 | 79.88 | 83.04 | 64.05 |
| LLMTTT | 88.53 | 86.22 | 79.67 | 80.02 | 73.82 | 90.63 | 88.05 | 84.89 | 86.19 | 75.62 |
| LLMGNN | 76.06 | 67.02 | 77.33 | 62.43 | 62.15 | 81.57 | 82.23 | 70.46 | 66.36 | 64.38 |

**Results.** As illustrated in Table 4, since LLMGNN is label-free, LLMGNN is not as effective as GCN trained on the training set. LLMTTT achieves far better results than GCN in the OOD test set, due to the additional information gain brought by LLMs. LLMGNN is a novel label-free node classification method that utilizes LLMs for node annotation to assist in the model training process. The performance of LLMGNN significantly surpasses that of other label-free node classification methods. Although the performance of LLMGNN may not match that of directly using LLMs for prediction, the costs are substantially lower compared to employing LLMs as predictors. LLMTTT, a variant of LLMGNN, incorporates LLMs as annotators during the test phase to generate pseudo-labels for fine-tuning the pre-trained model, resulting in improved performance compared to LLMGNN due to the integration of pre-trained models. Nevertheless, both LLMGNN and LLMTTT encounter two key technical challenges: the selection of a candidate node set based on multiple criteria, and the improvement of the accuracy of LLM annotations. Comprehensive results concerning LLMs as feature-level enhancer are in Appendix G.

**Observation 3:** *The application of LLMs as annotators represents a promising pipeline for addressing OOD challenge.*

### 4.3 LLMs-as-Predictors

**Experiment Implementation.** To answer **Q4**, we conduct experiments on two pipelines of using LLMs as predictors based on whether the LLM parameters can be updated. The LLMs whose parameters cannot be accessed or updated are denoted as online LLMs, and that can be deployed and finetuned offline as offline LLMs. For the online LLMs, the GPT-3.5-turbo-0613 model is adopted. We investigated various prompts, including prompts solely consisting of node features and that include both node features and structure information. Table 10 provides different design of prompts used in our experiment. More prompts used in online LLMs are detailed in Appendix D.1. For offline LLMs, we adopt InstructGLM-Llama-v1-7b (He et al., 2023b), which is finetuned on graph datasets via instruct tuning based on a series of scalable prompts especially designed to describe graphs.

Table 5: The results of online LLMs as predictor. Different prompts are used to guide LLM to generate prediction results on the OOD test set. "GCN" is the performance (%) of a well-trained GCN model. "Avg Rank" is the average ranking of this method among all the methods across various datasets. For each kind of prompt, we use yellow to denote the best performance under a specific GCN model, green the second best one, and blue the third one.

| | concept_degree | | | | | concept_word | | | | |
| --- | --- | --- | --- | --- | --- | --- | --- | --- | --- | --- |
| | Cora | Pubmed | CiteSeer | Wikics | Avg Rank | Cora | Pubmed | CiteSeer | Wikics | Avg Rank |
| GCN | 86.50 | 80.32 | 68.60 | 72.67 | 2.25 | 87.48 | 87.34 | 79.88 | 83.04 | 1.25 |
| Zero-shot | 64.40 | 87.84 | 60.92 | 66.02 | 3 | 61.94 | 89.54 | 54.16 | 50.98 | 2.75 |
| Few-shot | 67.03 | 91.23 | 74.03 | 65.15 | 2 | 54.97 | 79.82 | 56.02 | 65.00 | 3.75 |
| Zero-shot with 2-hop info | 60.45 | 89.32 | 53.48 | 61.55 | 4 | 60.02 | 79.46 | 47.08 | 60.10 | 4.5 |
| Few-shot with 2-hop info | 68.10 | 81.35 | 54.43 | 55.05 | 3.75 | 60.77 | 82.58 | 55.40 | 57.36 | 2.75 |

**Results.** As demonstrated in Table 5, large language models (LLMs) display markedly disparate performance across various datasets. In particular, they are outperformed by GCNs on the Cora dataset, while demonstrating superior performance on the PubMed dataset. While LLMs exhibit enhanced performance on certain datasets. On the whole, there is no significant and consistent results demonstrating that LLMs are better than GCN on the OOD challenge as predictors. Particularly, GCN shows better generalization performance than LLMs as predictors on the distribution shift concerning node features, which demonstrates that the message passing mechanism in GNNs can potentially alleviate challenge of feature shift among nodes.

The experiments demonstrated that even minor synonym variations in prompting can result in notable alterations to the prediction outcomes. This suggests that minor alterations to the prompts can markedly improve the performance of the model. Based on the Table 5, the introduction of few-shot samples only provides a minimal degree of assistance in the mitigation of annotation bias. In addition, the integration of structural information does not yield the anticipated performance improvements. In line with previous findings, the influence of prompts on LLM outcomes is considerable, underscoring the pivotal role of the approach used to incorporate neighbourhood information in TAGs.

In our experiments, the approach employed was based on the best-performing summary of neighbour information reported in prior research. However, the results indicate that there is scope for further improvement in prompt design. Given the substantial influence of prompts on the predictive performance of large models, the development of more effective prompts that incorporate structural information is a crucial area of research. This remains one of the major challenges in applying large models within the graph domain.

Due to the high cost associated with making predictions using LLMs, it's impractical to conduct multiple experiments. Meanwhile, significant variance in predictions has been

Table 6: The results of offline LLMs as predictor. "GCN" is the performance (%) of a well-trained GCN model. "Zero-shot" is the performance (%) of online LLM with zero-shot prompt. "InstructGLM" is the performance (%) of offline LLM model. "Avg Rank" is the average ranking of this method among all the methods across various datasets.

| | concept_degree | | | concept_word | | |
| --- | --- | --- | --- | --- | --- | --- |
| | Cora | Pubmed | Avg Rank | Cora | Pubmed | Avg Rank |
| GCN | 86.50 | 80.32 | 2 | 87.48 | 87.34 | 2 |
| Zero-shot | 64.40 | 87.84 | 2.5 | 61.94 | 89.54 | 2.5 |
| InstructGLM | 85.78 | 93.97 | 1.5 | 85.16 | 93.56 | 1.5 |

Table 7: The comparison of LLM cost used in different pipelines. "OOD acc" refers to the accuracy (%) of the corresponding method on the OOD test set under concept degree shift. "Cost" is obtained by converting the number of tokens into the corresponding cost in dollars. "Avg Rank" is the average ranking of this method among all the methods across various datasets. For each kind of methods, we use yellow to denote the best performance, green the second best one, and blue the third one.

| Pipeline | method | OOD acc | | | Cost | | |
|---|---|---|---|---|---|---|---|
| | | Cora | Pubmed | Avg Rank | Cora | Pubmed | Avg Rank |
| | GCN (sbert) | 86.50 | 80.32 | 4 | / | / | / |
| Enhancer | Ada | 86.98 | 80.98 | 3 | 0.07 | 0.75 | 1.5 |
| | TA+E | 84.23 | 84.24 | 4.5 | 1.33 | 13.93 | 4 |
| Predictor | Zero-shot | 86.50 | 80.32 | 4.5 | 0.32 | 4.00 | 3 |
| | InstructGLM | 85.78 | 93.97 | 3 | / | / | / |
| Annotator | LLMTTT | 88.53 | 86.22 | 1.5 | 0.07 | 0.81 | 1.5 |

observed in our experiments, with differences reaching up to 10% even under fixed parameters. Such unstable prediction results present considerable challenges for subsequent applications of LLMs and pose difficulties for the reproducibility of methods based on LLMs.

According to Table 6, InstructGLM demonstrates overall better performance than both the online LLM predictor and the GCN model, with a significantly superior performance on the PubMed OOD test set. In both online and offline LLM models, the constraints on token length imposed by LLMs result in an inability to adequately encompass all higher-order neighbours. There are currently two perspectives to address this issue: one approach involves the design of prompts that effectively incorporate structural information, while the other entails the construction of multiple descriptive sentences containing graph data to enumerate all possible neighbours at each hop. However, this latter approach can bring up a significant increase in the volume of training data. Overall, the development of reasonable and effective natural language description methods for graph structures represents a significant challenge in the current application of LLMs within the field of graph learning. Comprehensive results concerning LLMs as feature-level enhancer are in Appendix H.

**Observation 4:** *LLMs demonstrate promising generalization ability as predictors after fine-tuning, while well-designed prompts that can efficiently capture the graph information are essential.*

## 4.4 COMPARISON AMONG DIFFERENT LLM PIPELINES

Table 7 shows the comparison among the representative methods that achieve relatively good performance in different pipelines. From the perspective of generalization performance, the pipeline of LLM as annotators exhibit the most promising performance, with the pipeline of offline LLMs as predictor achieving the second best performance. Considering the expenditure brought by LLM usage, the pipelines of online LLMs as feature enhancer and as annotators are the most economical choices. Overall, LLMs can help alleviate the OOD challenge in TAGs, and to use LLMs as annotators is a promising direction, which achieves satisfactory improvements with affordable expenditure.

## 5 OUTLOOK

In this paper, we embark on a systematic study over the effect of LLMs towards OOD challenge on TAGs. We first develop an OOD dataset on five popular TAG datasets, and then examine three LLMs pipelines on the dataset from various perspectives. Our study reveals that LLMs can help alleviate OOD problem on TAGs if used properly. Specifically, to use LLMs as annotators or to leverage LLMs as predictors after fine-tuning are both promising predictions with affordable economic cost. Also, the design of prompts especially for graphs are essential for the effect of LLMs on TAGs. In addition, TF-IDF, a traditional node embedding method, show surprisingly good generalization performance. In the future, we plan to include more OOD types on a broad scope of TAGs into the developed OOD-TAG, and to incorporate more graph tasks into our bechmark, including graph classification task and link prediction task.

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

# A    DATASETS

We introduce covariate and concept shifts and devise data splits that precisely capture various shifts. More specific details in Table 8.

## A.1    TEXT-ATTRIBUTED GRAPH DATASETS

**CORA** dataset comprises 2708 scientific publications classified into seven classes, which is denoted as a small-scale citation network graph, where nodes represent scientific publications and edges are citation links. Each paper in this dataset has a least one citation connection with other paper.

**PUBMED** dataset consist of 19717 scientific journals from PubMed database pertaining to diabetes classified into three classes. The citation network consists of 44,338 links.

**CITESEER** dataset is a paper citation dataset with seven categories. The raw text attributes in the file only contain the text attributes for 3186 nodes. As a result, we take the graph consisted of these 3186 nodes with 4277 edges.

**WIKICS** dataset is a subset of Wikipedia-based dataset , which consists of 10 classes closely related to computer science. We get the raw attributes from raw dataset. The task is to predict the category of this article, given the contents of the Wikipedia article.

**ARXIV** dataset is a citation dataset adapted from OGB (Hu et al., 2020a). The input is a directed graph representing the citation network among the computer science (CS) arXiv papers indexed by MAG (Wang et al., 2020). Nodes in the graph represent arXiv papers, and directed edges represent citations. The task is predicting the subject area of arXiv CS papers, forming a 40-class classification problem.

Table 8: The details of the OOD-TAG datasets. There are five datasets in total, and four OOD principles are adopted to split these datasets. "con" denotes concept and "cova" represents covariate. Note that the number of test nodes from Pubmed and Citeseer is zero, since the degree distribution of them is relatively concentrated, it cannot be used as a covariate to divide the dataset.

| Dataset | Class | Num of nodes | Num of test nodes | | | |
|---------|-------|--------------|------------|---------------|-------------|---------------|
| | | | con_degree | con_word/time | cova_degree | cova_word/time |
| Cora | 7 | 2708 | 837 | 775 | 471 | 542 |
| Pubmed | 3 | 19717 | 6001 | 6193 | 0 | 3944 |
| CiteSeer | 6 | 3186 | 847 | 805 | 0 | 638 |
| Wikics | 10 | 11701 | 3308 | 3391 | 1740 | 2341 |
| ArXiv | 40 | 169343 | 51482 | 47910 | 20604 | 48603 |

## A.2    OUT-OF-DISTRIBUTION TAGS

Distribution shift in machine learning is generally divided into two types: covariate shift and concept shift (LEARNING; Moreno-Torres et al., 2012). Machine learning models typically predict an output $Y \in \mathcal{Y}$ based on an input $X \in \mathcal{X}$. The joint distribution $P(X, Y)$ between inputs and outputs can be written as $P(X, Y) = P(Y|X)P(X)$, which constitutes of two important distribution components.

**Covariate Shifts.** The assumption is that the covariate distribution $P(X)$ shifts while the concept distribution $P(Y|X)$ remains unchanged. The variable $X$ is related with multiple factors, and covariate shifts occur only in those factors that are unrelated to $Y$. Formally, a dataset can be viewed as a collection of $|\mathcal{D}|$ domain, where the distribution of each domain can be represented as $P_{d_i}(X)$. When a domain changes, for example, transitioning from domain $d_1$ to domain $d_2$, $P(Y|X)$ remains constant while $P(X)$ changes. For instance, in the ColoredMNIST dataset (Chen et al., 2022), the shape of the digit is the true determining factor for the digit category, whereas the color of the digit is an unrelated factor. Covariate shift refers to the phenomenon in which the digit's color changes without affecting the digit's category.

**Concept Shifts.** In contrast to covariate shift, concept shift takes into account the variation of the concept distribution $P(Y|X)$ across splits. The variable $X$ is composed of multiple factors, including

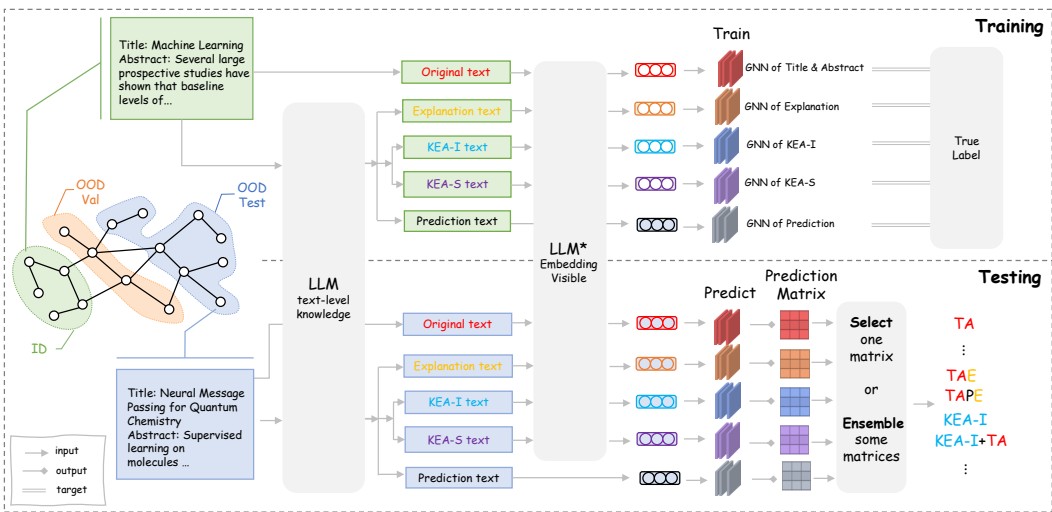

Figure 3: The pipelines for LLMs as enhancers. It is designed with two parts: (1) *Feature-level enhancement*, which injects LLMs' knowledge by encoding text attributes into features. Embedding-visible LLMs enhance text attributes directly by encoding them into initial node features for GNNs. (2) *Text-level enhancement,* which injects LLMs' knowledge by augmenting the text attributes at the text level. Embedding-invisible LLMs are initially adopted to enhance the text attributes by generating augmented attributes. The augmented attributes and original attributes are encoded and then ensembled together.

components that are related or unrelated to $Y$. Since some factors can completely determine $Y$, the relationship between these factors and $Y$ is invariant. So the concept distribution $P(Y|X)$ shift can only occur for those variables that are independent of $Y$. This means that these factors, which are not related to $Y$, create a spurious correlation between $Y$ and $X$. This spurious correlation is subsequently used to define various concepts, each of which can be viewed as a split. Formally, a dataset can be considered a mixture of $|\mathcal{C}|$ concepts, where the conditional distribution $P(Y|X)$ of each concept $c_k$ consists of multiple domains, with $P_{c_k}(Y \mid X = \mathbf{x_i}) = \sum_{j=1}^{|\mathcal{Y}|} q_{i,j}^k P_{y_j, d_i}(Y)$ represented for each domain $d_i$. Note that domain is a variable independent of $Y$. For instance, we divided the handwritten digit dataset ColoredMNIST into different concepts as described. In each concept, spurious correlations are generated between color domains and specific labels, which determine distinct concepts. Specifically, in concept $c_1$, green digits 90% represent the digit "1", while in concept $c_2$, green digits 30% correspond to the digit "1". In this case, different spurious correlations represent different concepts and cause different splits.

# B    LLMs as Enhancers

In the pipeline of *LLMs as enhancers*, LLMs are used to enhance the node features. As illustrated in Figure 3, there are two mainstream ways to leverage LLMs to enrich the text attributes, *i.e.*, feature-level enhancement and text-level enhancement (Chen et al., 2023). The feature-level enhancement utilizes LLMs' knowledge by encoding text attributes in to features. The text-level enhancement injects the LLMs' knowledge by augmenting the text attributes via extended text information.

## B.1    Feature-level Enhancement

For the feature-level enhancement, the LLMs work as feature embedding models whose outputs are then fed into small-scale backbone models, such as GNNs. As illustrated in Figure 3, only embedding-visible LLMs are taken into account to encode the text information.

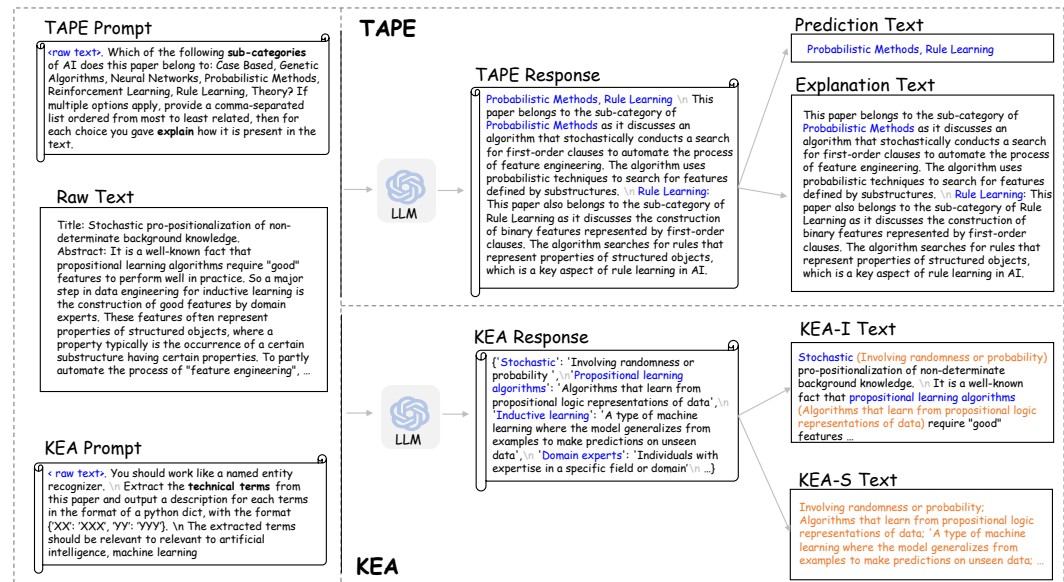

Figure 4: Illustrations of TAPE and KEA to enhance text attributes. TAPE leverages the knowledge of LLMs to generate explanations for their predictions. In the case of KEA, the LLMs are prompted to produce a list of technical terms accompanied by their descriptions.

### B.2 TEXT-LEVEL ENHANCEMENT

For the text-level enhancement, embedding-invisible LLMs are used to enrich the text attributes via various prompting strategies. Recently, several papers (He et al., 2023b; Chen et al., 2024) have explored the text-level enhancement in different manners. Here, we take TAPE (He et al., 2023b) and KEA (Chen et al., 2024) as baseline methods and the illustrative examples of these two augmentations are shown in Figure 4.

(1) TAPE (He et al., 2023b): Besides titles and abstracts, the TAPE extracts LLMs' knowledges by asking for predictions and their explanations. Given a piece of raw text, TAPE uses LLMs to directly make its prediction and asks the LLM to give corresponding explanation. By generating explanation, the TAPE edifies the LLMs to exploit the logical relationship between the text features and corresponding labels. For example, as shown in Figure 4, after making the predictions "Probabilistic Methods, Rule Learning", LLMs also provide the reasons for these predictions respectively. The prediction texts and the explanation texts are then encoded into embeddings, separately.

(2) KEA (Chen et al., 2024): The knowledge-enhanced augmentation (KEA) aims to explicitly incorporate external knowledge by prompting the LLMs to generate a list of knowledge entities along with their text descriptions. As illustrated in Figure 4, given one piece of raw text, the LLMs are asked to extract and explain the technical terms related to artificial intelligence and machine learning, instead of making direct prediction. There were two manners to use the descriptions of these knowledge entities. The KEA-I make the descriptions follow the corresponding technical terms in the raw texts, while the KEA-S simply piled the descriptions of these knowledge entities. The enhanced texts are then encoded into input features via embedding-visible LLMs, as illustrated in Figure 3. For each kind of input features, a GNN is trained specifically. During the test phase, we can choose any GNN or ensemble some of them to make predictions.

## C    LLMS AS ANNOTATORS

To provide a more comprehensive survey of various LLM pipelines, we examine two approaches that utilise LLMs as annotators and evaluate their performance on the TAGs OOD dataset. Figure 5 shows the pipelines of LLMTTT and LLMGNN. Further, we make the following observations from the experiments of LLMTTT and LLMGNN:

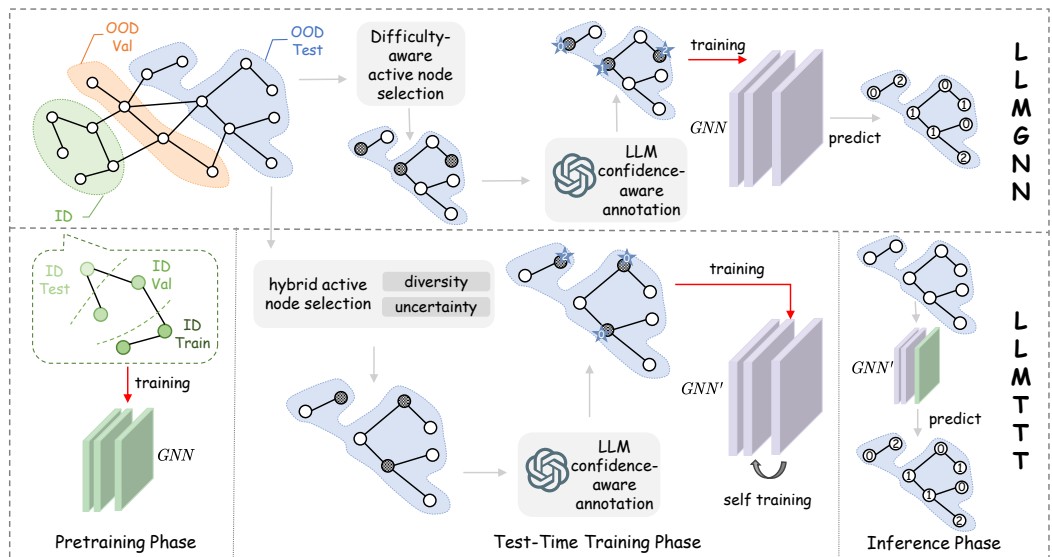

Figure 5: The pipelines for LLMs used as annotator in graphs. *LLMGNN*, which is designed for label-free node classification on graphs. It is designed with four components: (1) difficulty-aware active node selection. (2) confidence-aware annotations. (3) post-filtering. (4) GNN model training and prediction. *LLMTTT*, which performs test-time adaptation based on annotations provided by LLMs for a carefully-selected node set. It is designed with three components: (1)hybrid active node selection. (2) confidence-aware high-quality annotation. (3) two-stage training.

**Observation 5:** *The selection of the annotated node set presents a significant challenge.*

Given the high cost of prediction with LLMs, it is necessary to generate pseudo labels for only a subset of nodes, which requires the prior identification of a candidate node set. The selection of this set should be based on confidence levels, which can be estimated from the amount of additional information available; this constitutes a critical technical challenge when LLMs are utilized as annotators. Directly using LLM outputs to assess confidence levels is not feasible, as LLMs often provide outputs that lack meaningful interpretation. Instead, LLMs that support prediction logits can be employed to evaluate node confidence based on the output probabilities.

Moreover, nodes with the same annotation confidence may have varying impacts on other nodes within the graph. Annotating certain nodes can lead to greater performance improvements, making the identification of key nodes within the dataset essential. LLMGNN integrates traditional active learning with post-filtering strategies for node selection and demonstrates the effectiveness of the post-filtering approach. Additionally, LLMTTT leverages the predictive signals provided by the pre-trained model to enhance node selection, with experimental results indicating that incorporating a greater amount of information indeed yields significant performance gains.

**Observation 6:** *The accuracy of annotations by LLMs determines the ceiling of model performance.*

Intuitively, the classification performance of the model is closely related to the precision of LLM annotations. We validate this hypothesis in LLMTTT. After establishing a fixed candidate node set, the accuracy of the selected nodes' labels was deliberately controlled. It was observed that following the fixation of the candidate node set, the accuracy of pseudo-labels determined the ceiling of the model's classification performance. Both LLMTTT and LLMGNN utilized a confidence-based node annotation method when employing LLMs for node annotation. However, experimental results indicated that the accuracy of the pseudo-labels generated by our prompts did not meet expectations, impacting the subsequent classification performance of the model. Therefore, future research should focus on developing a method for selecting more reliable nodes for LLMs.

Given that subsequent model fine-tuning relies on annotations from LLMs, further exploration of the factors significantly influencing model fine-tuning performance is warranted. Using LLMTTT as an example, both LLM label accuracy (LLM acc) and GCN accuracy (GCN acc) were controlled

for the selected nodes to assess whether the impact of TTT results was solely determined by label accuracy. Pseudo-labels assigned to the selected nodes were treated as true labels, incorporating varying degrees of perturbation. Results presented in the Table 4 indicate that, although the pseudo-label accuracy of nodes used for fine-tuning was identical, the final performance (TTT acc) varied due to differences in performance gaps between GCN and LLM. A larger gap corresponded to higher classification performance. This suggests that the accuracy of TTT is not solely positively correlated with pseudo-label accuracy but may be highly related to the performance disparity between GCN and LLM. Thus, addressing the bucket effect is essential. In other words, when selecting nodes, attention should be directed toward nodes with high predictive entropy from the pre-trained GNN model.

**Observation 7:** *The accuracy of LLM annotations and the selection of node sets mutually influence each other, necessitating a balanced approach based on practical considerations.*

Utilizing LLMs as annotators presents two challenges that collectively impact overall performance. Taking LLMGNN as an example, although the C-Density active node selection strategy can yield higher annotation quality, relying solely on this metric may lead to suboptimal model training performance. This is due to the candidate node set determined by this active selection strategy, which introduces issues of label imbalance. Therefore, careful consideration must be given to both annotation accuracy and the candidate node set in practical situations, as balancing these factors can lead to improved model performance.

# D    LLMs AS PREDICTORS

Figure 6 shows the two pipelines of LLMs used as predictor in graphs.

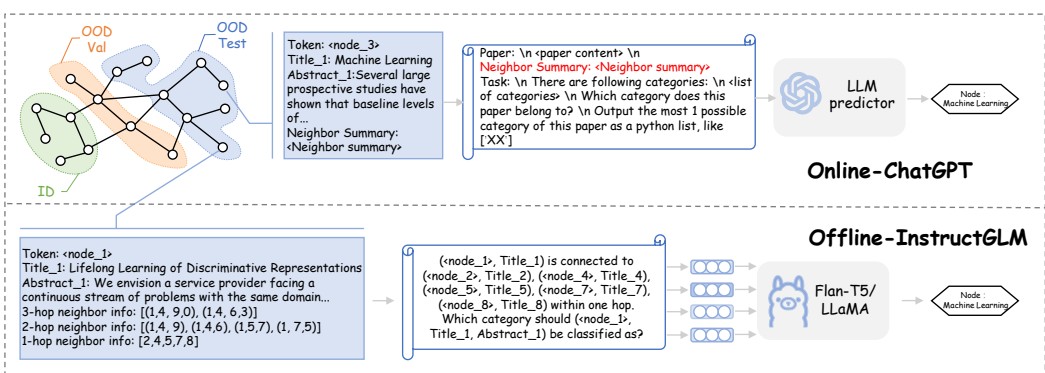

Figure 6: The pipelines for LLMs used as predictor in graphs. It is designed with two pipelines: (1) *Online ChatGPT*, where LLMs directly make the predictions. The key component for this pipeline is how to design an effective prompt to incorporate structural and attribute information. (2) *Offline InstructGLM*, which utilizes natural language to describe both graph structure and meta features of node and edge for generative LLMs and further addressing graph-related tasks by instruction-tuning.

## D.1    PROMPTS USED IN LLMs AS PREDICTOR.

The crucial element of LLMs-as-predictor is the design of prompts. Table 10 provides different design of prompts used in our experiment. Each prompt comprises the paper's title and abstract, followed by a task-related question. The answer section is left blank for the model to complete. Overall, answers are derived from the text output of LLMs. To evaluate the impact of different prompts, we also incorporated structural information from various perspectives into the prompt. In particular, the prompts in the Table 9 are employed to request that LLMs generate a neighbour information summary of the current node, based on neighbour features and labels. This design of prompt simulates the neighbour information aggregation process of GNNs. Given that GNNs are predominantly composed of two layers, the information aggregation of two-hop neighbours is considered in this experiment. Given that LLMs have context limitations, the neighbours of the current node are sampled once, and information is only summarised for the selected neighbours.

Table 9: Prompts used to generate neighbor summary.

---

The following list records some papers related to the current one.
[{ "content": "Cadabra a field theory motivated ...", "category": "computer vision"... }, ...]
Please summarize the information above with a short paragraph, find some common points which can reflect the category of this paper

---

Table 10: The design of prompts used in LLMs as predictors. Each prompt comprises the paper's title and abstract, followed by a task-related question. The answer section is left blank for the model to complete. Overall, answers are derived from the text output of LLMs. To evaluate the impact of different prompts, we also incorporated structural information from various perspectives into the prompt. In particular, the prompts in the Table 9 are employed to request the LLMs to generate a neighbour information summary of the current node. This design of prompt simulates the neighbour information aggregation process of GNNs. Given that GNNs are predominantly composed of two layers, the information aggregation of two-hop neighbours is considered in this experiment. Given that LLMs have context limitations, the neighbours of the current node are sampled once, and information is only summarised for the selected neighbours.

| prompt name | prompt content |
|---|---|
| Zero-shot | Paper: \n <paper content>\n 
 Task: \n There are following categories: \n <list of categories>\n What's the category of this paper Output your answer together with a confidence ranging from 0 to 100, in the form of a list of python dicts like [{"answer":<answer_here>, "confidence": <confidence_here>}] |
| Few-shot | # Information for the first few-shot samples \n 
 Paper: \n <paper content>\n 
 Task: \n There are following categories: \n <list of categories>\n What's the category of this paper Output your answer together with a confidence ranging from 0 to 100, in the form of a list of python dicts like [{"answer":<answer_here>, "confidence": <confidence_here>}] |
| Zero-shot with 2-hop info | Paper: \n <paper content>\n 
 Neighbor Summary: <Neighbor summary>\n 
 Task: \n There are following categories: \n <list of categories>\n What's the category of this paper Output your answer together with a confidence ranging from 0 to 100, in the form of a list of python dicts like [{"answer":<answer_here>, "confidence": <confidence_here>}] |
| Fero-shot with 2-hop info | # Information for the first few-shot samples \n 
 Paper: \n <paper content>\n 
 Neighbor Summary: <Neighbor summary>\n 
 Task: \n There are following categories: \n <list of categories>\n What's the category of this paper Output your answer together with a confidence ranging from 0 to 100, in the form of a list of python dicts like [{"answer":<answer_here>, "confidence": <confidence_here>}] |

## E    COMPLETE RESULTS OF LLMS AS FEATURE-LEVEL ENHANCERS.

This section includes the complete results of LLMs as feature-level enhancers under covariate and concept shifts in GCN and MLP model.

The complete results of LLMs as feature-level enhancers under concept degree shift in GCN model are shown in Table 1, which in MLP model are shown in Table 11.

The complete results of LLMs as feature-level enhancers under concept word shift in GCN model are shown in Table 2, which in MLP model are shown in Table 12.

The complete results of LLMs as feature-level enhancers under covariate degree shift in GCN model are shown in Table 13, which in MLP model are shown in Table 15.

The complete results of LLMs as feature-level enhancers under covariate word shift in GCN model are shown in Table 14, which in MLP model are shown in Table 16.

The performance gap of GCN and MLP with different embeddings on ID and OOD test data under various distribution shifts is shown in Figure 7.

Table 11: The results of LLM as feature-level enhancers for concept degree shift. Different MLP models are trained with different embeddings on the ID training set. "ID" denotes the GCN performance (%) on the ID test set, and "OOD" denotes the GCN performance (%) on the OOD test set. "Avg Rank" is the average ranking of this method among all the methods across various datasets. For each kind of features, we use yellow to denote the best performance under a specific MLP model, green the second best one, and blue the third one.

| | Cora | | Pubmed | | CiteSeer | | Wikics | | ArXiv | | Avg Rank | |
|---|---|---|---|---|---|---|---|---|---|---|---|---|
| | ID | OOD | ID | OOD | ID | OOD | ID | OOD | ID | OOD | ID | OOD |
| *Non-contextualized Shallow Embeddings* | | | | | | | | | | | | |
| TF-IDF | 78.51 | 76.25 | 85.17 | 79.71 | 73.54 | 64.13 | 77.07 | 74.39 | 69.36 | 59.65 | 4 | 3.8 |
| Word2Vec | 62.89 | 63.94 | 78.85 | 70.81 | 64.31 | 61.11 | 75.17 | 69.96 | 57.14 | 46.82 | 5.6 | 5.4 |
| *Local Word Embedding models* | | | | | | | | | | | | |
| DeBERTa | 35.02 | 34.93 | 67.71 | 56.38 | 49.54 | 37.33 | 45.31 | 32.95 | 39.37 | 27.5 | 7 | 7 |
| LLaMa | 65.07 | 59.47 | 75.73 | 65.02 | 60.31 | 53.34 | 83.77 | 81.15 | 74.3 | 65.14 | 4.2 | 4.4 |
| *Local Sentence Embedding Models* | | | | | | | | | | | | |
| SBERT | 80.6 | 80.43 | 87.38 | 85.39 | 73.62 | 67.63 | 77.53 | 71.23 | 74.56 | 65.67 | 2.2 | 2.6 |
| E5 | 75.82 | 75.79 | 86.72 | 85.77 | 73.23 | 68.15 | 78.17 | 73.05 | 74.24 | 65.23 | 3.4 | 2.8 |
| *Online Sentence Embedding Model* | | | | | | | | | | | | |
| Ada | 82.79 | 80.05 | 90.94 | 88.58 | 76.77 | 69.82 | 84.09 | 79.44 | 72.95 | 63.23 | 1.6 | 2 |

Table 12: The results of LLM as feature-level enhancers for concept word shift. Different MLP models are trained with different embeddings on the ID training set. "ID" denotes the GCN performance (%) on the ID test set, and "OOD" denotes the GCN performance (%) on the OOD test set. "Avg Rank" is the average ranking of this method among all the methods across various datasets. For each kind of features, we use yellow to denote the best performance under a specific MLP model, green the second best one, and blue the third one.

| | Cora | | Pubmed | | CiteSeer | | Wikics | | ArXiv | | Avg Rank | |
|---|---|---|---|---|---|---|---|---|---|---|---|---|
| | ID | OOD | ID | OOD | ID | OOD | ID | OOD | ID | OOD | ID | OOD |
| *Non-contextualized Shallow Embeddings* | | | | | | | | | | | | |
| TF-IDF | 82.16 | 79.91 | 84.15 | 84.80 | 70.48 | 72.45 | 79.41 | 75.25 | 68.57 | 62.73 | 3.8 | 3.2 |
| Word2Vec | 68.05 | 63.18 | 74.03 | 76.39 | 63.69 | 67.92 | 74.94 | 71.84 | 56.27 | 51.88 | 5.8 | 5.4 |
| *Local Word Embedding models* | | | | | | | | | | | | |
| DeBERTa | 53.25 | 31.00 | 64.59 | 61.31 | 50.77 | 39.85 | 45.91 | 29.96 | 41.46 | 25.11 | 7 | 7 |
| LLaMa | 69.35 | 59.15 | 74.87 | 72.34 | 54.46 | 54.59 | 82.47 | 79.61 | 73.75 | 64.57 | 4.2 | 4.2 |
| *Local Sentence Embedding Models* | | | | | | | | | | | | |
| SBERT | 85.89 | 80.51 | 87.41 | 84.87 | 74.10 | 75.79 | 79.01 | 74.84 | 75.38 | 58.67 | 2.2 | 2.8 |
| E5 | 79.48 | 72.31 | 86.14 | 86.09 | 74.39 | 75.41 | 78.24 | 74.90 | 75.76 | 60.26 | 2.8 | 3.4 |
| *Online Sentence Embedding Model* | | | | | | | | | | | | |
| Ada | 84.33 | 79.10 | 89.82 | 89.59 | 72.18 | 73.33 | 83.95 | 81.07 | 72.83 | 64.28 | 2.2 | 2 |

Table 13: The results of LLM as feature-level enhancers for covariate degree shift. Different GCN models are trained with different embeddings on the ID training set. "ID" denotes the GCN performance (%) on the ID test set, and "OOD" denotes the GCN performance (%) on the OOD test set. "Avg Rank" is the average ranking of this method among all the methods across various datasets. For each kind of features, we use yellow to denote the best performance under a specific GCN model, green the second best one, and blue the third one.

| | Cora | | Pubmed | | CiteSeer | | Wikics | | ArXiv | | Avg Rank | |
|---|---|---|---|---|---|---|---|---|---|---|---|---|
| | ID | OOD | ID | OOD | ID | OOD | ID | OOD | ID | OOD | ID | OOD |
| *Non-contextualized Shallow Embeddings* | | | | | | | | | | | | |
| TF-IDF | 87.33 | 83.10 | / | / | / | / | 83.85 | 64.98 | 77.42 | 58.50 | 1.33 | 5 |
| Word2Vec | 85.48 | 76.94 | / | / | / | / | 81.86 | 71.93 | 76.01 | 60.39 | 5.67 | 4.67 |
| *Local Word Embedding models* | | | | | | | | | | | | |
| DeBERTa | 83.56 | 69.04 | / | / | / | / | 76.03 | 56.07 | 58.94 | 27.59 | 7 | 7 |
| LLaMa | 86.30 | 78.98 | / | / | / | / | 81.38 | 73.32 | 76.51 | 65.48 | 5.33 | 3 |
| *Local Sentence Embedding Models* | | | | | | | | | | | | |
| SBERT | 87.19 | 82.63 | / | / | / | / | 83.42 | 68.78 | 76.85 | 63.17 | 2.67 | 4 |
| E5 | 86.59 | 83.57 | / | / | / | / | 82.91 | 67.26 | 77.49 | 65.94 | 2.67 | 2.33 |
| *Online Sentence Embedding Model* | | | | | | | | | | | | |
| Ada | 86.81 | 83.44 | / | / | / | / | 82.27 | 79.90 | 77.05 | 63.83 | 3.33 | 2 |

Table 14: The results of LLM as feature-level enhancers for covariate word shift. Different GCN models are trained with different embeddings on the ID training set. "ID" denotes the GCN performance (%) on the ID test set, and "OOD" denotes the GCN performance (%) on the OOD test set. "Avg Rank" is the average ranking of this method among all the methods across various datasets. For each kind of features, we use yellow to denote the best performance under a specific GCN model, green the second best one, and blue the third one.

| | Cora | | Pubmed | | CiteSeer | | Wikics | | ArXiv | | Avg Rank | |
|---|---|---|---|---|---|---|---|---|---|---|---|---|
| | ID | OOD | ID | OOD | ID | OOD | ID | OOD | ID | OOD | ID | OOD |
| *Non-contextualized Shallow Embeddings* | | | | | | | | | | | | |
| TF-IDF | 87.85 | 85.65 | 87.08 | 86.62 | 71.32 | 71.00 | 83.30 | 81.48 | 75.40 | 72.37 | 3.6 | 3.8 |
| Word2Vec | 84.59 | 81.44 | 85.22 | 84.70 | 69.69 | 65.58 | 82.43 | 80.37 | 73.75 | 70.76 | 5.2 | 5.8 |
| *Local Word Embedding models* | | | | | | | | | | | | |
| DeBERTa | 82.81 | 82.10 | 83.93 | 83.64 | 53.46 | 43.64 | 74.32 | 67.23 | 56.08 | 52.96 | 6.8 | 6.6 |
| LLaMa | 83.04 | 82.10 | 86.47 | 85.00 | 71.70 | 76.39 | 81.59 | 81.61 | 75.28 | 72.86 | 4.8 | 3.2 |
| *Local Sentence Embedding Models* | | | | | | | | | | | | |
| SBERT | 88.67 | 88.63 | 87.27 | 86.16 | 75.22 | 74.76 | 83.45 | 80.98 | 59.01 | 56.84 | 3 | 3.8 |
| E5 | 87.56 | 86.42 | 89.06 | 86.93 | 74.53 | 74.36 | 83.73 | 81.45 | 75.91 | 72.98 | 1.8 | 2.8 |
| *Online Sentence Embedding Model* | | | | | | | | | | | | |
| Ada | 88.52 | 87.56 | 88.51 | 87.21 | 76.42 | 78.90 | 83.30 | 82.50 | 75.63 | 72.74 | 2 | 1.6 |

Table 15: The results of LLM as feature-level enhancers for covariate degree shift. Different MLP models are trained with different embeddings on the ID training set. "ID" denotes the GCN performance (%) on the ID test set, and "OOD" denotes the GCN performance (%) on the OOD test set. "Avg Rank" is the average ranking of this method among all the methods across various datasets. For each kind of features, we use yellow to denote the best performance under a specific MLP model, green the second best one, and blue the third one.

| | Cora | | Pubmed | | CiteSeer | | Wikics | | ArXiv | | Avg Rank | |
|---|---|---|---|---|---|---|---|---|---|---|---|---|
| | ID | OOD | ID | OOD | ID | OOD | ID | OOD | ID | OOD | ID | OOD |
| *Non-contextualized Shallow Embeddings* | | | | | | | | | | | | |
| TF-IDF | 82.22 | 74.95 | / | / | / | / | 80.51 | 68.34 | 70.99 | 56.41 | 3 | 4 |
| Word2Vec | 67.11 | 64.97 | / | / | / | / | 75.21 | 66.07 | 58.95 | 41.16 | 5.67 | 5.67 |
| *Local Word Embedding models* | | | | | | | | | | | | |
| DeBERTa | 43.41 | 40.55 | / | / | / | / | 45.56 | 30.45 | 39.21 | 19.76 | 7 | 7 |
| LLaMa | 61.70 | 59.41 | / | / | / | / | 84.14 | 75.45 | 74.76 | 65.16 | 3 | 2.67 |
| *Local Sentence Embedding Models* | | | | | | | | | | | | |
| SBERT | 80.15 | 80.55 | / | / | / | / | 80.32 | 67.00 | 74.08 | 64.34 | 3.33 | 3 |
| E5 | 75.78 | 73.67 | / | / | / | / | 80.36 | 69.80 | 74.24 | 64.40 | 3.33 | 3 |
| *Online Sentence Embedding Model* | | | | | | | | | | | | |
| Ada | 79.41 | 77.92 | / | / | / | / | 85.01 | 75.22 | 73.03 | 63.19 | 2.67 | 2.67 |

Table 16: The results of LLM as feature-level enhancers for covariate word shift. Different MLP models are trained with different embeddings on the ID training set. "ID" denotes the GCN performance (%) on the ID test set, and "OOD" denotes the GCN performance (%) on the OOD test set. "Avg Rank" is the average ranking of this method among all the methods across various datasets. For each kind of features, we use yellow to denote the best performance under a specific MLP model, green the second best one, and blue the third one.

| | Cora | | Pubmed | | CiteSeer | | Wikics | | ArXiv | | Avg Rank | |
|---|---|---|---|---|---|---|---|---|---|---|---|---|
| | ID | OOD | ID | OOD | ID | OOD | ID | OOD | ID | OOD | ID | OOD |
| *Non-contextualized Shallow Embeddings* | | | | | | | | | | | | |
| TF-IDF | 80.07 | 67.56 | 85.31 | 83.19 | 70.88 | 69.12 | 80.22 | 74.02 | 66.76 | 65.94 | 4 | 4 |
| Word2Vec | 70.52 | 53.43 | 76.13 | 73.27 | 69.50 | 65.58 | 75.06 | 69.60 | 54.83 | 54.96 | 5.6 | 5.6 |
| *Local Word Embedding models* | | | | | | | | | | | | |
| DeBERTa | 42.89 | 32.21 | 58.90 | 50.53 | 45.16 | 35.61 | 45.20 | 31.91 | 33.52 | 32.72 | 7 | 7 |
| LLaMa | 67.70 | 41.55 | 78.38 | 79.48 | 53.21 | 56.83 | 82.34 | 80.81 | 72.39 | 71.27 | 4 | 4 |
| *Local Sentence Embedding Models* | | | | | | | | | | | | |
| SBERT | 81.48 | 72.69 | 85.77 | 85.57 | 71.82 | 73.13 | 80.89 | 77.44 | 71.76 | 69.39 | 2.6 | 2.4 |
| E5 | 80.37 | 65.42 | 81.19 | 80.39 | 73.52 | 73.26 | 78.14 | 77.10 | 72.16 | 69.65 | 3.2 | 3.2 |
| *Online Sentence Embedding Model* | | | | | | | | | | | | |
| Ada | 83.56 | 72.21 | 88.92 | 88.42 | 73.90 | 75.27 | 83.93 | 82.53 | 70.73 | 68.89 | 1.6 | 1.8 |

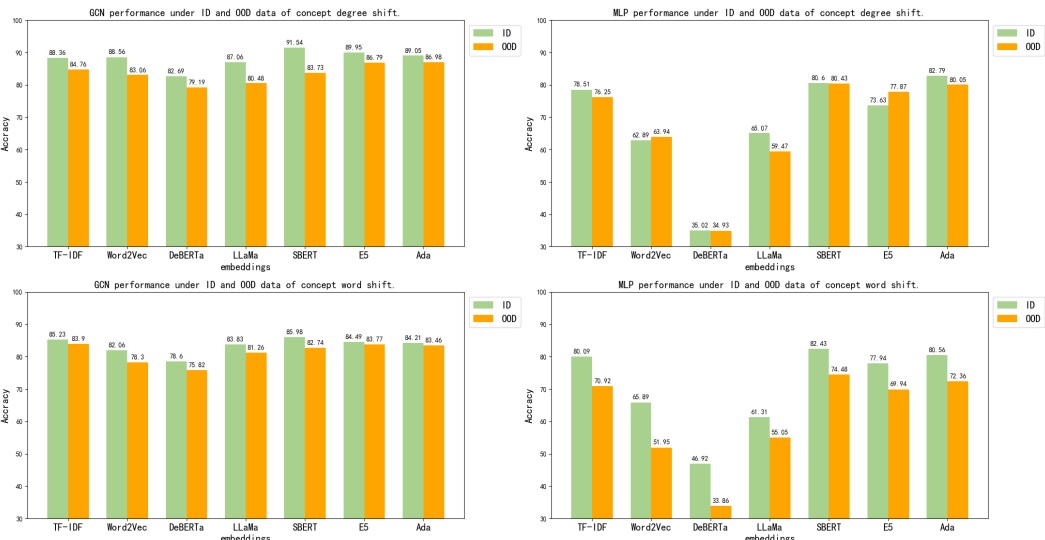

Figure 7: The performance gap of GCN and MLP with different embeddings on ID and OOD test data under various distribution shifts.

## F COMPLETE RESULTS OF LLMS AS TEXT-LEVEL ENHANCERS.

This section includes the complete results of LLMs as text-level enhancers under covariate and concept shifts in GCN and MLP model.

The complete results of LLMs as text-level enhancers under concept shift in MLP model are shown in Table 17.

The complete results of LLMs as text-level enhancers under concept shift in GCN model are shown in Table 18, which in MLP model are shown in Table 19.

Table 17: The results of LLM as text-level enhancers. Different MLP models are trained with different texts on the ID training set. "ID" denotes the MLP performance (%) on the ID test set, and "OOD" denotes the MLP performance (%) on the OOD test set. "Avg Rank" is the average ranking of this method among all the methods across various datasets. "Pseudo Label" is the prediction accuracy coming from the direct prediction from LLMs during the TAPE process. For each kind of methods, we use yellow to denote the best performance, green the second best one, and blue the third one.

| Concept Degree | Cora | | Pubmed | | Avg Rank | | Concept Word | Cora | | Pubmed | | Avg Rank | |
|---|---|---|---|---|---|---|---|---|---|---|---|---|---|
| | ID | OOD | ID | OOD | ID | OOD | | ID | OOD | ID | OOD | ID | OOD |
| TA(e5) | 74.73 | 77.54 | 86.38 | 81.91 | 6.5 | 5.5 | TA(e5) | 77.20 | 68.00 | 85.45 | 85.63 | 6.5 | 6.5 |
| Pseudo Label | 66.17 | 66.55 | 94.66 | 93.52 | 5 | 5.5 | Pseudo Label | 67.85 | 64.13 | 93.58 | 93.58 | 5.5 | 6 |
| P | 56.52 | 58.35 | 94.66 | 93.55 | 5.5 | 5.5 | P | 61.68 | 54.92 | 93.47 | 93.59 | 6.5 | 6 |
| E | 77.31 | 74.48 | 93.14 | 91.91 | 4.5 | 5.5 | E | 75.23 | 69.08 | 92.27 | 92.73 | 6 | 4.5 |
| TA+E | 80.70 | 79.98 | 93.00 | 91.11 | 3 | 3.5 | TA+E | 82.06 | 74.61 | 92.83 | 92.46 | 2.5 | 3 |
| TA+P+E | 80.50 | 79.16 | 94.17 | 92.83 | 2.5 | 3 | TA+P+E | 81.50 | 74.40 | 93.60 | 93.70 | 1.5 | 1.5 |
| KEA-I | 76.22 | 75.03 | 83.79 | 78.41 | 7.5 | 7 | KEA-I | 77.94 | 67.92 | 84.33 | 85.53 | 6.5 | 7.5 |
| KEA-S | 68.76 | 72.40 | 77.18 | 67.97 | 9 | 9 | KEA-S | 71.31 | 75.41 | 76.47 | 9 | 9 |
| KEA-I+TA | 78.91 | 79.14 | 86.26 | 81.07 | 5 | 5.5 | KEA-I+TA | 79.07 | 69.91 | 85.67 | 86.32 | 4.5 | 5 |
| KEA-S+TA | 78.11 | 80.91 | 84.11 | 77.59 | 6 | 5 | KEA-S+TA | 78.88 | 71.20 | 84.07 | 84.66 | 6.5 | 6 |

Table 18: The results of LLM as text-level enhancers. Different GCN models are trained with different texts on the ID training set. "ID" denotes the GCN performance (%) on the ID test set, and "OOD" denotes the GCN performance (%) on the OOD test set. "Avg Rank" is the average ranking of this method among all the methods across various datasets. "Pseudo Label" is the prediction accuracy coming from the direct prediction from LLMs during the TAPE process. For each kind of methods, we use yellow to denote the best performance, green the second best one, and blue the third one.

| Covariate Degree | Cora ID | Cora OOD | Avg Rank ID | Avg Rank OOD | Covariate Word | Cora ID | Cora OOD | Pubmed ID | Pubmed OOD | Avg Rank ID | Avg Rank OOD |
|---|---|---|---|---|---|---|---|---|---|---|---|
| TA(E5) | 86.00 | 81.83 | 6 | 5 | TA | 85.56 | 87.05 | 87.17 | 85.24 | 6 | 5 |
| Pseudo Label | 59.26 | 65.82 | 10 | 10 | Pseudo Label | 67.78 | 64.21 | 94.47 | 91.55 | 5 | 5 |
| P | 61.78 | 52.1 | 9 | 9 | P | 61.7 | 56.42 | 77.95 | 80.49 | 10 | 10 |
| E | 86.44 | 79.62 | 5 | 7 | E | 83.19 | 84.39 | 91.21 | 89.66 | 5 | 4 |
| TA+E | 87.04 | 82.08 | 3 | 3 | TA+E | 85.78 | 86.61 | 90.09 | 88.62 | 2.5 | 3.5 |
| TA+P+E | 86.59 | 82.04 | 4 | 4 | TA+P+E | 84.30 | 86.49 | 90.30 | 88.85 | 4.5 | 3.5 |
| KEA-I | 88.15 | 85.05 | 1 | 1 | KEA-I | 84.81 | 83.36 | 88.58 | 86.32 | 5.5 | 7.5 |
| KEA-S | 84.67 | 74.31 | 8 | 8 | KEA-S | 83.85 | 83.87 | 85.65 | 84.32 | 8 | 8 |
| KEA-I+TA | 87.70 | 83.95 | 2 | 2 | KEA-I+TA | 85.63 | 86.16 | 88.66 | 86.78 | 4 | 5 |
| KEA-S+TA | 85.56 | 80.59 | 7 | 6 | KEA-S+TA | 85.7 | 87.31 | 88.19 | 86.47 | 4.5 | 3.5 |

Table 19: The results of LLM as text-level enhancers. Different MLP models are trained with different texts on the ID training set. "ID" denotes the MLP performance (%) on the ID test set, and "OOD" denotes the MLP performance (%) on the OOD test set. "Avg Rank" is the average ranking of this method among all the methods across various datasets. "Pseudo Label" is the prediction accuracy coming from the direct prediction from LLMs during the TAPE process. For each kind of methods, we use yellow to denote the best performance, green the second best one, and blue the third one.

| Covariate Degree | Cora ID | Cora OOD | Avg Rank ID | Avg Rank OOD | Covariate Word | Cora ID | Cora OOD | Pubmed ID | Pubmed OOD | Avg Rank ID | Avg Rank OOD |
|---|---|---|---|---|---|---|---|---|---|---|---|
| TA | 74.81 | 74.10 | 5 | 6 | TA | 78.96 | 73.65 | 76.85 | 71.96 | 7.5 | 8 |
| Pseudo Label | 59.26 | 65.82 | 9 | 9 | Pseudo Label | 67.78 | 64.21 | 94.47 | 91.55 | 5.5 | 5.5 |
| P | 54.37 | 59.32 | 10 | 10 | P | 62.89 | 54.28 | 94.82 | 91.63 | 5.5 | 5.5 |
| E | 70.67 | 71.51 | 8 | 8 | E | 78.81 | 74.69 | 93.35 | 90.83 | 5 | 4.5 |
| TA+E | 76.59 | 76.18 | 3 | 4 | TA+E | 80.22 | 77.53 | 90.53 | 88.44 | 4.5 | 3 |
| TA+P+E | 75.04 | 76.69 | 4 | 3 | TA+P+E | 81.26 | 76.57 | 93.93 | 91.25 | 2.5 | 3.5 |
| KEA-I | 73.63 | 74.52 | 6 | 5 | KEA-I | 78.81 | 69.37 | 83.63 | 80.25 | 7 | 6.5 |
| KEA-S | 72.44 | 71.80 | 7 | 7 | KEA-S | 74.30 | 67.68 | 79.10 | 75.37 | 7.5 | 8.5 |
| KEA-I+TA | 78.07 | 77.37 | 2 | 2 | KEA-I+TA | 82.15 | 76.90 | 81.35 | 77.75 | 4 | 4.5 |
| KEA-S+TA | 78.67 | 77.45 | 1 | 1 | KEA-S+TA | 80.67 | 76.61 | 79.95 | 76.13 | 5.5 | 5.5 |

# G    COMPLETE RESULTS OF LLMs AS ANNOTATORS.

This section includes the complete results of LLMs as annotators under covariate shifts.

The complete results of online LLMs as annotators under coavariate shift are shown in Table 20.

Table 20: The results (%) of LLM as annotators. "LLMTTT" and "LLMGNN" are two different pipeline in which LLMs are used as annotators.

| | covariate_degree | | | | | covariate_word | | | | |
|---|---|---|---|---|---|---|---|---|---|---|
| | Cora | Pubmed | CiteSeer | Wikics | ArXiv | Cora | Pubmed | CiteSeer | Wikics | ArXiv |
| GCN | 87.05 | / | / | 71.72 | 61.21 | 90.41 | 86.56 | 79.47 | 81.76 | 69.98 |
| LLMTTT | 91.08 | / | / | 86.35 | 75.06 | 92.25 | 86.97 | 86.33 | 86.35 | 75.06 |
| LLMGNN | 62.46 | / | / | 78.46 | 59.85 | 82.18 | 82.18 | 74.51 | 74.17 | 65.26 |

# H COMPLETE RESULTS OF LLMs AS PREDICTORS.

This section includes the complete results of LLMs as Predictors under covariate shifts.

The complete results of online LLMs as predictors under coavariate shift are shown in Table 21.

The complete results of online LLMs as predictors under coavariate shift are shown in Table 22.

Table 21: The results of online LLMs as predictors. Different prompts are used to guide LLM to generate prediction results on the OOD test set. "GCN" is the performance (%) of a well-trained GCN model. "Avg Rank" is the average ranking of this method among all the methods across various datasets. For each kind of prompt, we use yellow to denote the best performance under a specific GCN model, green the second best one, and blue the third one.

| | covariate_degree | | | | | covariate_word | | | | |
|---|---|---|---|---|---|---|---|---|---|---|
| | Cora | Pubmed | CiteSeer | Wikics | Avg Rank | Cora | Pubmed | CiteSeer | Wikics | Avg Rank |
| GCN | 87.05 | / | / | 71.72 | 1 | 90.41 | 86.56 | 79.47 | 81.76 | 1.25 |
| Zero-shot | 55.81 | / | / | 57.36 | 3 | 54.80 | 87.30 | 57.68 | 55.97 | 2.25 |
| Few-shot | 58.17 | / | / | 55.11 | 2.5 | 52.58 | 81.16 | 63.32 | 54.93 | 3.25 |
| Zero-shot with 2-hop info | 57.75 | / | / | 48.97 | 3.5 | 53.69 | 81.87 | 55.33 | 51.90 | 3.75 |
| Few-shot with 2-hop info | 47.56 | / | / | 45.52 | 5 | 45.94 | 80.48 | 57.84 | 50.06 | 4.5 |

Table 22: The results of offline LLMs as predictors. "GCN" is the result (%) of a well-trained GCN model. "Zero-shot" is the result (%) of online LLM with zero-shot prompt. "InstructGLM" is the result (%) of offline LLM model. "Avg Rank" is the average ranking of this method among all the methods across various datasets.

| | covariate_degree | | covariate_word | | |
|---|---|---|---|---|---|
| | Cora | Avg Rank | Cora | Pubmed | Avg Rank |
| GCN | 87.05 | 2 | 90.41 | 86.56 | 2 |
| Zero-shot | 55.81 | 5 | 54.80 | 87.30 | 2.5 |
| InstructGLM | 87.26 | 1 | 87.45 | 94.17 | 1.5 |

