# OpenReview forum: "Towards the Effect of Large Language Models on Out-Of-Distribution Challenge in Text-Attributed Graphs"
_ICLR.cc/2025/Conference — ICLR 2025 Conference Withdrawn Submission_

### Official Review · Reviewer_e1Dz · 2024-10-31

**Soundness:** 2
**Presentation:** 2
**Contribution:** 1
**Rating:** 3
**Confidence:** 3

**Summary:**

This paper introduces OOD-TAG, a benchmark on the generalization capability of three types of LLM-compatible pipelines in text-attributed graphs. In this paper, benchmark datasets with diverse distribution shifts are developed. To evaluate the ability of LLM-based methods on generalizing OOD TAGs, this study conducts extensive experiments and conclude several findings.

**Strengths:**

- The research topic, LLM-based OOD generalization on TAGs, is important.
- The paper conducts extensive experiments and provides the code repository, ensuring high reproducibility.
- The paper analyze three paradigm that augments node classification tasks on TAGs with LLMs.

**Weaknesses:**

- Baslines and LLMs adopted in this paper are a little bit out-of-date.
- There is room for improvement in the writing.
- Detailed information of datasets are unavailable. For example, the number of domains or environments in each dataset and the statistics of different splits (e.g. training , ID validation, ID test, OOD validation, and OOD test).
- The paper conducts extensive analysis of different types of LLM-based methods, but it lacks perspectives and experiments specifically addressing OOD issues.

**Questions:**

- What are the differences between datasets in this study and those with the same names (e.g. Cora, ArXiv) from the GOOD benchmark?
- It is confusing that only performance on OOD test set are given in Section4.2 and 4.3.
- Are the GCNs in Tables 11 and 12 actually meant to be MLPs?

---

### Official Review · Reviewer_hno8 · 2024-11-04

**Soundness:** 3
**Presentation:** 4
**Contribution:** 3
**Rating:** 3
**Confidence:** 5

**Summary:**

This paper addresses the OOD challenge in TAGs. To support this research, the authors first develop an OOD dataset based on five popular TAG datasets and then evaluate three LLM pipelines on this dataset from various perspectives. Extensive experiments across diverse evaluation settings benchmark the performance of 16 methods from these pipelines, offering insights into when and how LLMs can assist in addressing OOD in the graph domain.

**Strengths:**

- This paper introduces a novel reformulation of the standard graph OOD problem in the context of graph LLMs.
- The authors aim to provide a comprehensive evaluation of LLM-compatible pipelines in the graph domain from three perspectives: LLM-as-Enhancers, LLM-as-Annotators, and LLM-as-Predictors.
- Extensive experiments are conducted to assess the performance of various graph LLM pipelines under out-of-distribution shift scenarios. Additionally, a comparative analysis among the three pipelines offers a general understanding of the different design approaches.

**Weaknesses:**

- The statement about developing an OOD dataset on TAGs seems somewhat overstated. When I first read the title, I expected the authors to provide new definitions of OOD specific to TAGs. However, the primary OOD construction process closely follows GOOD without emphasizing the differences between TAGs and standard graphs. For instance, ogbn-Arxiv could be easily viewed as an OOD dataset in GOOD by converting textual attributes into numerical features, as seen in the OGB leaderboard. Including word diversity as a new criterion for TAGs is reasonable, but the impact of textual attributes on the OOD problem in TAGs versus standard graphs should be explicitly discussed and clarified.
- This does not represent a comprehensive OOD study on TAGs, as it excludes recent advances in the graph LLM field. There have been numerous recent developments in this area, which should be discussed or included in the OOD evaluation setup. For example, in Table 1, the feature embedding baselines considered are not representative, as they are not tailored for TAGs; examples include DeBERTa, Sentence-BERT, TF-IDF, and Word2Vec. Incorporating textual attributes into graph embeddings has become a prominent research area, and a few leading works should be included, such as PATTON [1], GAINT [2], and UniGLM [3]. For more comprehensive references, see https://github.com/PeterGriffinJin/Awesome-Language-Model-on-Graphs.
- Similarly, the selection of comparative methods for using LLMs as predictors is rather limited and lacks representative models. Notable methods include GraphGPT[4], LLaGA[5], and GraphTranslator[6], which are well-recognized and already published in the literature.
- The paper lacks an adequate discussion of related works. Three widely recognized research directions in the graph field are LLM-as-Enhancer, LLM-as-Annotator, and LLM-as-Predictor. Numerous efforts have been made to push the performance boundaries in these areas, and these methods should be discussed at a minimum. Key works include [7][8], and more comprehensive references are available at https://github.com/PeterGriffinJin/Awesome-Language-Model-on-Graphs.
- The diversity of evaluation datasets is limited in terms of domain. Many popular TAG benchmarks are available online, such as the Amaon-X datasets from the e-commerce field, which may exhibit different graph characteristics from citation networks. To make the insights more robust, TAG datasets from various domains should be included in the experiments.

[1] Patton: Language model pretraining on text-rich networks.

[2] Node feature extraction by selfsupervised multi-scale neighborhood prediction.

[3] UniGLM: Training One Unified Language Model for Text-Attributed Graphs.

[4] Graph Instruction Tuning for Large Language Models

[5] LLaGA: Large Language and Graph Assistant

[6] GraphTranslator: Aligning Graph Model to Large Language Model for Open-ended Tasks

[7] UniGraph: Learning a Unified Cross-Domain Foundation Model for Text-Attributed Graphs

[8] GAugLLM: Improving Graph Contrastive Learning for Text-attributed Graphs with Large Language Models

**Questions:**

I appreciate the research problem addressed in this work. However, a more comprehensive evaluation, a thorough discussion of related works, and a clearer illustration of the OOD differences between TAGs and standard graphs are needed. For further details, please refer to the Weaknesses section.

---

### Official Review · Reviewer_esM9 · 2024-11-04

**Soundness:** 3
**Presentation:** 2
**Contribution:** 3
**Rating:** 5
**Confidence:** 3

**Summary:**

The out-of-distribution(OOD) challenge in the text-attributed graphs(TAGs) domain is a classic research problem. To address this issue, this paper pioneers the construction of a benchmark for evaluating the effect of LLMs on tackling the OOD challenges in TAGs. This benchmark consists of five typical publicly available academic citation TAGs, which are detailedly categorized according to specific OOD types. Based on this benchmark, the experiments presented in the paper assess multiple representative works categorized from three perspectives: LLM as enhancer, annotator, and predictor, and yield some insightful conclusions.

**Strengths:**

1. The OOD challenge in the textual graph domain is a classic and widespread research problem, making technical evaluation in this area of significant practical importance.
2. As the primary contribution, the benchmark constructed in this work is comprehensive, innovative, and highly valuable in practical applications.

**Weaknesses:**

1. All the data in the benchmark dataset comes from academic citation networks, which is somewhat limited. In fact, OOD tasks are also common in TAGs from other domains (e.g., social network TAG like Reddit [https://convokit.cornell.edu/documentation/subreddit.html]). Thus, the benchmark could be further refined from the perspectives of different domains or textual content.

2. The paper lacks sufficient background research. Several significant and well-known TAG+LLM methods, such as [1], [2], [3], and [4], have not been discussed. Additionally, some benchmark papers on TAG, like [5], are missing from the discussion. The authors should consider these methods and explain why they were not included in the experiments (if need). Furthermore, the differences between this paper and current TAG benchmarks should also be consider.

**Reference**


   [1] Ziwei Chai, Tianjie Zhang, Liang Wu, Kaiqiao Han, Xiaohai Hu, Xuanwen Huang, and Yang Yang. "Graphllm: Boosting graph reasoning ability of large language model." arXiv preprint arXiv:2310.05845 (2023).

   [2] Jiabin Tang, Yuhao Yang, Wei Wei, Lei Shi, Lixin Su, Suqi Cheng, Dawei Yin, and Chao Huang. "Graphgpt: Graph instruction tuning for large language models." In Proceedings of the 47th International ACM SIGIR Conference on Research and Development in Information Retrieval, pp. 491-500. 2024.

   [3] Xuanwen Huang, Kaiqiao Han, Yang Yang, Dezheng Bao, Quanjin Tao, Ziwei Chai, and Qi Zhu. "Can GNN be Good Adapter for LLMs?." In Proceedings of the ACM on Web Conference 2024, pp. 893-904. 2024.

   [4] Zirui Guo, Lianghao Xia, Yanhua Yu, Yuling Wang, Zixuan Yang, Wei Wei, Liang Pang, Tat-Seng Chua, and Chao Huang. "Graphedit: Large language models for graph structure learning." arXiv preprint arXiv:2402.15183 (2024).

   [5] Yuhan Li, Peisong Wang, Xiao Zhu, Aochuan Chen, Haiyun Jiang, Deng Cai, Victor Wai Kin Chan, and Jia Li. "GLBench: A Comprehensive Benchmark for Graph with Large Language Models." arXiv preprint arXiv:2407.07457 (2024)

**Questions:**

See Weaknesses.

---

### Official Review · Reviewer_5hdi · 2024-11-06

**Soundness:** 2
**Presentation:** 4
**Contribution:** 2
**Rating:** 3
**Confidence:** 5

**Summary:**

This paper aims at providing benchmark to out-of-distribution problems for text-attributed graphs. The authors provides four OOD graph datasets based on different train/validation/test splits of existing graph datasets. Then different categories of methods such as LLM as enhancer, LLM as predictor, and LLM as annotator are benchmarked on the data.

**Strengths:**

1. Studying OOD problem on TAG is an interesting research direction.
2. Presentation is clear, the writing of this paper is easy to follow.
3. Experimental part is comprehensive based on the give datasets.

**Weaknesses:**

1. The OOD problem on TAG is not well defined. The authors only give very high-level idea that the distribution shift exists between training and test datasets. However, no in-depth definition was given. More importantly, as this paper focus on text-attributed graphs, the definition of OOD in this paper seems has no direct relation with text. The examples of splits are based on node degree and time, which requires in-depth explanation on why this is related to TAGs.

2. Although there are many experimental results in the paper, they are mostly focus on studying the performance difference of different categories of LLM+GNN methods, which are not directly related to the problem. For experimental results, important results lacks such as when the distribution shift becomes larger, how the model prediction ability will be affected.

3. Even though this is a benchmark paper, some promising protocol solutions to address the OOD issues for TAGs are still desired.

**Questions:**

Are there some quantitative metrics can be utilized to measure the distributional shift?

Can authors split the data based on text distributions?

---

### Note · Authors · 2024-12-04

I have read and agree with the venue's withdrawal policy on behalf of myself and my co-authors.